



**Real-time measurements of gas-phase organic acids using SF₆⁻ chemical ionization**
**mass spectrometry**
Theodora Nah,[1] Yi Ji,[1,2] David J. Tanner,[1] Hongyu Guo,[1] Amy P. Sullivan,[3] Nga Lee Ng,[1,2]
Rodney J. Weber[1] and L. Gregory Huey[1*]
[1]School of Earth and Atmospheric Sciences, Georgia Institute of Technology, Atlanta, GA, USA
[2]School of Chemical and Biomolecular Engineering, Georgia Institute of Technology, Atlanta, GA, USA
[3]Department of Atmospheric Science, Colorado State University, Fort Collins, CO, USA
* To whom correspondence should be addressed: greg.huey@eas.gatech.edu
**Abstract**
The sources and atmospheric chemistry of gas-phase organic acids are currently poorly
understood due in part to the limited range of measurement techniques available. In this
work, we evaluated the use of $SF_6^-$ as a sensitive and selective chemical ionization reagent
ion for real-time measurements of gas-phase organic acids. Field measurements are made
using a chemical ionization mass spectrometer (CIMS) at a rural site in Yorkville, Georgia
from September to October 2016 to investigate the capability of this measurement
technique. Our measurements demonstrate that $SF_6^-$ can be used to measure a range of
organic acids in the atmosphere. Ambient concentrations of organic acids ranged from a
few parts per trillion by volume (ppt) to several parts per billion by volume (ppb).
Assuming that these organic acids are completely water-soluble, the carbon mass fraction
of gas-phase water-soluble organic carbon ($WSOC_g$) comprised of these organic acids
ranged from 7 to 100 % with a study average of 30 %. All the organic acids displayed
similar strong diurnal behaviors, reaching maximum concentrations between 5 and 7 pm
local time. The organic acid concentrations are dependent on ambient temperature, with
higher organic acid concentrations being measured during warmer periods.
**Introduction**
Organic acids are ubiquitous and important species in the troposphere. They are
major contributors of free acidity in precipitation (Galloway et al., 1982; Keene et al., 1983;
Keene and Galloway, 1984), and can also affect the formation of secondary organic
aerosols (SOA) (Zhang et al., 2004; Carlton et al., 2006; Sorooshian et al., 2010; Yatavelli




et al., 2015). As end products of oxidation, organic acids can also serve as useful tracers of
air mass history (Sorooshian et al., 2007; Sorooshian et al., 2010). Organic acids are found
in urban, rural and remote marine environments in the gas, aqueous and particle phases.
While organic acids are emitted directly from biogenic sources (e.g., microbial activity,
vegetation and soil) and anthropogenic activities (e.g., fossil fuel combustion, vehicular
emissions and biomass burning) (Kawamura et al., 1985; Talbot et al., 1988; Chebbi and
Carlier, 1996; Talbot et al., 1999; Seco et al., 2007; Veres et al., 2010; Paulot et al., 2011;
Veres et al., 2011; Millet et al., 2015), they can also be formed from photooxidation of
non-methane volatile organic compounds and aqueous-phase photochemistry of semi-
volatile organic compounds (Chebbi and Carlier, 1996; Hansen et al., 2003; Orzechowska
and Paulson, 2005; Carlton et al., 2006; Sorooshian et al., 2007; Ervens et al., 2008; Paulot
et al., 2011; Millet et al., 2015). The chemical aging of organic aerosols has also been
proposed as a major source of organic acids (Paulot et al., 2011). The relative importance
of primary and secondary sources of organic acids are currently poorly constrained though
their emissions likely depend on the magnitude of biogenic and anthropogenic activities
and the meteorological conditions. Wet and dry deposition are the primary sinks of organic
acids in the atmosphere (Chebbi and Carlier, 1996).

Formic and acetic acids are the dominant gas-phase monocarboxylic acids in the

troposphere (Chebbi and Carlier, 1996). Due to their high vapor pressures, the gas-phase
concentrations of formic and acetic acids are usually 1 to 2 orders of magnitudes higher
than their particle-phase concentrations. Some field studies report strong correlations
between formic and acetic acids, suggesting that these two organic acids have similar
sources (Nolte et al., 1997; Souza and Carvalho, 2001; Paulot et al., 2011). A recent
modeling study suggested that the dominant sources of formic acid in the southeastern U.S.
are primarily biogenic in nature (Millet et al., 2015). These sources include direct emissions
from vegetation and soil and photochemical production from biogenic volatile organic
compounds (BVOCs). Currently, atmospheric formic and acetic acid concentrations are
higher than those predicted by models, indicating that present model estimates of source
and sink magnitudes are incorrect (Paulot et al., 2011; Millet et al., 2015). In the case of
formic acid, deposition and secondary photochemical production via mechanisms such as
photooxidation of isoprene and reaction of stabilized criegee intermediates need to be



better constrained in models. Given that formic and acetic acids are major trace gases in
the atmosphere, there is a need to resolve the discrepancy between measurements and
model predictions to close the atmospheric reactive carbon budget and improve our overall
understanding of VOC chemistry in the atmosphere.

Currently, research on gas-phase organic acids has focused primarily on formic and

acetic acids (Andreae et al., 1988; Talbot et al., 1988; Grosjean, 1991; Hartmann et al.,
1991; Talbot et al., 1995; Talbot et al., 1999). This is due, in part, to the analytical
difficulties in measuring gas-phase $> C_2$ organic acids and oxidized organic acids (i.e.,
containing more than 2 oxygen atoms) in real time. These organic acids have low vapor
pressures and are generally present in low concentrations in the gas phase. For example,
dicarboxylic acids typically have vapor pressures that are 2 to 4 orders of magnitude lower
than their analogous monocarboxylic acids (Chebbi and Carlier, 1996), and are present
mainly in the particle and aqueous phases. Rapid and accurate measurements of gas-phase
$> C_2$ organic acids and oxidized organic acids are necessary for constraining the regional
and global SOA budget since these acids can partition readily between the gas and particle
and aqueous phases and subsequently affect SOA formation (Zhang et al., 2004; Carlton
et al., 2006; Ervens et al., 2008; Sorooshian et al., 2010; Yatavelli et al., 2015).

Chemical ionization mass spectrometry (CIMS) is commonly used to selectively

measure atmospheric trace gases in real-time with high sensitivity. CIMS measurements
rely on reactions between reagent ions and compounds of interest present in the sampled
air to produce analyte ions that are detected by a mass spectrometer. The subset of
molecular species detected is determined by the reagent ion employed since the specificity
of the ionization process is governed by the ion-molecule reaction mechanism. CIMS is a
popular tool for atmospheric measurements since it provides high time resolution, linear
and reproducible measurements. It is also a soft ionization technique with minimal ion
fragmentation, thus preserving the parent molecule's elemental composition and allowing
for molecular speciation. Recent developments in chemical ionization methods and sources
have greatly improved our ability to measure atmospheric acidic species. Some of the
CIMS reagent ions that have been used to measure atmospheric organic acids include
acetate ($CH_3CO_2^-$), iodide ($I^-$) and $CF_3O^-$ anions (Crounse et al., 2006; Veres et al., 2008;




Lee et al., 2014; Brophy and Farmer, 2015; Nguyen et al., 2015). However, each of these
CIMS reagent ions has its drawbacks, which are generally related to their selectivity and
sensitivity towards different atmospheric species. For example, acetic acid is difficult to
measure with $CH_3CO_2^-$ as the CIMS reagent ion due to interferences from the reagent ion
chemistry that complicates the desired ion-molecule reactions. In addition, while many
organic acids can be detected using $I^-$ as a reagent ion, its sensitivity to different acids can
vary by orders of magnitude (Lee et al., 2014). For these reasons, this work is focused on
assessing the ability of $SF_6^-$ to measure a series of organic acids in ambient air.

The sulfur hexafluoride ($SF_6^-$) anion has been used as a CIMS reagent ion to

measure atmospheric inorganic species such as sulfur dioxide ($SO_2$), nitric acid ($HNO_3$)
and peroxynitric acid ($HO_2NO_2$) (Slusher et al., 2001; Slusher et al., 2002; Huey et al.,
2004; Kim et al., 2007). $SF_6^-$ commonly reacts with most acidic gases at the collision rate
by either proton or fluoride transfer reactions (Huey et al., 1995). However, $SF_6^-$ is reactive
to both ozone ($O_3$) and water vapor, which can lead to interfering reactions that limit its
applicability to many species in certain environments (Huey et al., 2004). In this work, we
present ambient measurements of gas-phase organic acids conducted in a mixed forest-
agricultural area in Georgia in early fall of 2016 to evaluate the performance of a $SF_6^-$
CIMS technique. Gas-phase organic acid measurements are compared to gas-phase water-
soluble organic carbon ($WSOC_g$) measurements performed during the field study to
estimate the fraction of $WSOC_g$ that is comprised of organic acids at this rural site.
Laboratory experiments are conducted to measure the sensitivity of $SF_6^-$ with a series of
organic acids of atmospheric relevance.
**2. Methods**
**2.1. Field site**

Real-time ambient measurements of gas-phase organic acids were obtained using a

chemical ionization mass spectrometer from 3 Sept to 12 Oct 2016 at the SouthEastern
Aerosol Research and Characterization (SEARCH) site located in Yorkville, Georgia. A
detailed description of the field site has been provided by Hansen et al. (2003). Briefly, the
Yorkville field site (33.931 N, 85.046 W) was located ~55 km northwest of Atlanta, and
was on a broad ridge in a large pasture where there were occasionally grazing cattle. The
field site was surrounded by forest and agricultural land. There were no major roads near
the field site and nearby traffic emissions were negligible. The sampling period was
characterized by moderate temperatures (24.0 °C average, 32.6 °C max, 9.5 °C min) and
high relative humidities (68.9% RH average, 100% RH max, 21.6% RH min). The study-
averaged diurnal trends of relative humidity, temperature and solar radiance are shown in
Fig. S1. Data reported are displayed in EDT. Volumetric gas concentrations reported are
at ambient temperature and relative humidity.

**2.2. $SF_6^-$ CIMS**

**2.2.1. CIMS instrument and air sampling inlet**

The CIMS instrument was housed in a temperature controlled trailer during the

field study. The inlet configuration and CIMS instrument used in this study is shown in
Fig. 1. Since $HNO_3$ and organic acids may condense on surfaces, an inlet configuration
with a minimal wall interaction was used. This inlet configuration was previously described
by Huey et al. (2004) and Nowak et al. (2006); hence, only a brief description will be
provided here. The inlet was a 7.6 cm ID aluminum pipe that extended ~40 cm into the
ambient air through a hole in the trailer's wall. This positioned the inlet ~2 m above the
ground. A donut-shaped ring was attached to the ambient sampling port of the pipe to
reduce the influence of crosswinds on the pipe's flow dynamics. This ring was wrapped
with a fine wire mesh to prevent insects from being drawn through the pipe. A flow of
~2800 L min$^{-1}$ was maintained in the pipe using a regenerative blower (AMETEK
Windjammer 116637-03). Part of this flow (7 L min$^{-1}$) was sampled through a custom-
made three-way PFA Teflon valve, which connected the pipe's center to the CIMS
sampling orifice. The valve was maintained at a temperature of 40 °C in an insulated
aluminum oven and could be switched automatically between ambient and background
modes. In ambient mode, ambient air was passed through a 25 cm long, 0.65 cm ID Teflon
tube into the CIMS. In background mode, ambient air was first drawn through an activated
charcoal scrubber before being delivered into the CIMS. A small flow of ambient air
(~0.05 L min$^{-1}$) was continuously passed through the scrubber to keep it at equilibrium
with ambient humidity levels. Most of the sampled air flow (6.7 L min$^{-1}$) was exhausted



using a small diaphragm pump. The rest of the sampled air flow (0.3 L min$^{-1}$) was
introduced into the CIMS instrument through an automatic variable orifice, which was used
to maintain a constant sample air mass flow.
Detailed descriptions of the CIMS instrument can be found in Liao et al. (2011) and
Chen et al. (2016). Briefly, the CIMS instrument was comprised of a series of differentially
pumped regions: a flow tube, a collisional dissociation chamber, an octopole ion guide, a
quadrupole mass filter and an ion detector. These sections were evacuated by a scroll pump
(Varian Triscroll$^{TM}$ 300), a drag pump (Adixen MDP 5011) and two turbo pumps (Varian
Turbo-V70), respectively. Ambient air was drawn continuously into the flow tube. A flow
of 3.7 L min$^{-1}$ of $N_2$ containing a few ppm of $SF_6$ (Scott-Marrin Inc.) was passed through
a $^{210}$Po ion source into the flow tube. $SF_6^-$ anions, which were produced via associative
electron attachment in the $^{210}$Po ion source, reacted with the sampled ambient air in the
flow tube to generate analyte ions. The flow tube was maintained at a low pressure (~13
mbar) to minimize interferences from $SF_6^-$ reaction with water vapor. The analyte ions then
exited the flow tube and were accelerated through the collisional dissociation chamber
(CDC), which was maintained at ~0.8 mbar. The molecular collisions in the CDC served
to dissociate weakly bound cluster ions into their core ions to simplify mass spectral
analysis. For this study, the CDC was operated at a relatively high voltage (-50 V) to
efficiently dissociate cluster ions. The resulting ions were then passed into the octopole ion
guide (maintained at ~6 x 10$^{-3}$ mbar), which collimated the ions and transferred them into
the quadrupole mass spectrometer (maintained at ~10$^{-5}$ mbar) for mass selection and
detection.
**2.2.2. Background and calibration measurements during field study**
Background measurements were performed every 25 min during the field study.
During each background measurement, the sampled air flow was passed through an
activated charcoal scrubber prior to delivery into the CIMS. The scrubber removed > 99 %
of the targeted species in ambient air. Calibration measurements were performed every 5 h
during the field study through standard additions of $^{34}SO_2$ and either formic or acetic acid
to the sampled air flow. Each background and calibration measurement period lasted ~4
min. A 1.12 ppm $^{34}SO_2$ gas standard was used as the source of the sulfur standard addition.





The formic and acetic acid calibration sources were permeation tubes (VICI Metronics)
with emission rates of 91 and 110 ng min$^{-1}$, respectively. The emission rates were measured
by scrubbing the output of the permeation tube in deionized water via a gas impinge
immersed in water, which was then analyzed for formate and acetate using ion
chromatography (Thermo Fisher Scientific). Eight samples of each acid were analyzed
over the course of the field study and the standard deviations of the permeation rates were
$\leq 6\,\%$. The CIMS instrument sensitivity measured by the $F_2{}^{34}SO_2{}^{-}$ ion signal (m/z 104) was
similarly applied to all the other measured species (except for formic and acetic acids)
using relative sensitivities determined in laboratory studies. The $F_2{}^{34}SO_2{}^{-}$ calibrant ion
signals were also used to calibrate ambient $F_2{}^{32}SO_2{}^{-}$ ion signals and determine ambient $SO_2$
concentrations as discussed in section 3.2.5.
**2.2.3. Laboratory calibration**
$HNO_3$, oxalic, butyric, glycolic, propionic and valeric acid standard addition
calibrations were performed in post-field laboratory work. The response of the CIMS acid
signals were measured relative to the sensitivity of $^{34}SO_2$ in these calibration
measurements. The $HNO_3$ calibration source was a permeation tube (KIN-TEK) with a
permeation rate of 39 ng min$^{-1}$, which was measured using UV optical absorption (Neuman
et al., 2003). Solid or liquid samples of oxalic (Sigma Aldrich, $\geq 99\,\%$), butyric (Sigma
Aldrich, $\geq 99\,\%$), glycolic (Sigma Aldrich, 99 %), propionic (Sigma Aldrich, $\geq 99.5\,\%$)
and valeric (Sigma Aldrich, $\geq 99\,\%$) acids were used in calibration measurements. The acid
sample was placed in a glass impinger, which was immersed in an ice bath to provide a
constant vapor pressure. A flow of 6 to 10 mL min$^{-1}$ of $N_2$ was passed over the organic acid
in the glass impinger. This organic acid air stream was then diluted with varying flows of
$N_2$ (1 to 5 L min$^{-1}$) to achieve different mixing ratios of the organic acid. Mixing ratios
were calculated from either the acid's emission rate from the impinger or the acid's vapor
pressure. The emission rate of gas-phase oxalic acid from the impinger was measured by
scrubbing the output in deionized water using the same method for calibrating the formic
and acetic acid permeation tubes, followed by ion chromatography analysis for oxalate.
Three samples were analyzed and the emission rate was determined to be 14 ng min$^{-1}$ with
a standard deviation of < 5 %. The vapor pressures of butyric and propionic acids at 0 ˚C



were measured using a capacitance manometer (MKS Instruments). The vapor pressures
of glycolic and valeric acids at 0 ˚C were estimated using their literature vapor pressures
at 25 ˚C and enthalpies of vaporization (Daubert and Danner, 1989; Lide, 1995; Acree and
Chickos, 2010).

Attempts to generate a calibration plot for pyruvic acid using its liquid sample

(Sigma Aldrich, 98 %) and the setup described above were unsuccessful as this acid was
found to interact very strongly with surfaces. Glyoxylic acid calibrations were not
performed due to the presence of impurities in the glyoxylic acid monohydrate solution
used (Sigma Aldrich, 98 %), which resulted in the appearance of ions not attributed to
glyoxylic acid. We attempted to generate calibration plots for malonic (Sigma Aldrich, ≥
99.5 %), succinic (Sigma Aldrich, 99 %) and glutaric (Sigma Aldrich, 99 %) acids by
passing $N_2$ over their solid samples at room temperature. However, it was not possible to
generate large enough gas phase concentrations for calibration since these organic acids
have very low vapor pressures.

**2.2.4. Detection limits and measurement uncertainties**

The detection limits of the organic acids were approximated from 3 times the

standard deviation values (3σ) of the ion signals measured during background mode. Table
1 summarizes the average detection limits of the organic acids for 2.5 min integration
periods which corresponds to the length of a background measurement with a 0.04 s duty
cycle for each m/z. The mean difference between successive background measurements
ranged from 1 to 40 ppt for the different organic acids. Future work will focus on reducing
the instrument background, and therefore improving the detection limits of these organic
acids.

The uncertainties (1σ) in our ambient measurements of formic, acetic and oxalic

acid concentrations originated from CIMS and IC calibration measurements. The IC
measurement uncertainty was estimated to be 10 %. For formic and acetic acids, which
were calibrated during the field study using permeation tubes, their CIMS measurement
uncertainties were estimated to be 6 and 7 %, respectively, based on one standard deviation
of the acids' calibrant ion signals. For oxalic acid which was calibrated in post-field





laboratory work, the CIMS measurement uncertainty was estimated to be 9 % based on one
standard deviation of the $^{34}SO_2$ sensitivity (3 %), the acid's calibrant ion signals (7 %) and
linear fit of the calibration curve (5 %). Hence, the uncertainties in our ambient
measurements of formic, acetic and oxalic acid concentrations were estimated to be 12, 12
and 14 %, respectively.
For nitric acid which was calibrated in post-field laboratory work using a
permeation tube and UV optical absorption, the uncertainty in its ambient concentrations
was estimated to be 9 % based on uncertainties in UV absorption measurements (3 %) and
one standard deviation of the $^{34}SO_2$ sensitivity (3 %) and acid's calibrant ion signals (8 %).
For butyric and propionic acids which were calibrated in post-field laboratory work using
vapor pressures measured by a capacitance manometer, the uncertainties in their ambient
concentrations were estimated to be 14 % based on the vapor pressure measurement
uncertainty (10 %) and one standard deviation of the $^{34}SO_2$ sensitivity (3 %), the acids'
calibrant ion signals (8 %) and linear fits of the acids' calibration curves (3 %). For glycolic
and valeric acids which were calibrated in post-field laboratory work using vapor pressures
estimated from literature vapor pressures at 25 ˚C and enthalpies of vaporization, the
uncertainties in their ambient concentrations were likely significantly larger compared to
the other measured organic acids due to uncertainties in their estimated vapor pressures.
We estimate the uncertainties in ambient concentrations of glycolic and valeric acids to be
22 % based on an assumed vapor pressure uncertainty of 20 % and one standard deviation
of the $^{34}SO_2$ sensitivity (3 %), the acids' calibrant ion signals (8 %) and linear fits of the
acids' calibration curves (2 %).
**2.3. WSOC$_g$ measurements**
WSOC$_g$ was measured with a MIST chamber coupled to a total organic carbon
(TOC) analyzer (Sievers 900 series, GE Analytical Instruments). Ambient air first passed
through a Teflon filter (45 mm diameter, 2.0 μm pore size, Pall Life Sciences) to remove
particles in the air stream. This filter was changed every 3 to 4 days. The particle-free air
was then pulled into a glass Mist Chamber filled with ultrapure deionized water at a flow
rate of 20 L min$^{-1}$. The MIST chamber scrubbed soluble gases with Henry's law constants
greater than $10^3$ M atm$^{-1}$ into deionized water (Spaulding et al., 2002). The resulting liquid





samples from the MIST chamber were analyzed by the TOC analyzer. The TOC analyzer
converted the organic carbon in the liquid samples to carbon dioxide using UV light and
chemical oxidation. The carbon dioxide formed was then measured by conductivity. The
amount of organic carbon in the liquid samples is proportional to the measured increase in
conductivity of the dissolved carbon dioxide. Each $WSOC_g$ measurement lasted 4 min.
Background $WSOC_g$ measurements were performed for 45 min every 12 h by stopping the
sample air flow and rinsing the sampling lines with deionized water. The TOC analyzer
was calibrated using different concentrations of sucrose (as specified by the instrument
manual) before and after the field study. The limit of detection was 0.4 $\mu gC\ m^{-3}$. The
measurement uncertainty was estimated to be 10 % based on uncertainties in the sample
air flow, liquid flow and TOC analyzer uncertainty. The MIST chamber and upstream
particle filter were located in an air-conditioned building so were generally below ambient
temperature. Hence, evaporation of collected particles (which will lead to positive artifacts
in $WSOC_g$ measurements) are not expected to be significant.
**2.4. Supporting gas measurements**

Supporting gas measurements were provided by a suite of instruments operated by

the SEARCH network. A non-dispersive infrared spectrometer (Thermo Fisher Scientific)
provided hourly CO measurements. A UV absorption analyzer (Thermo Fisher Scientific)
provided hourly $O_3$ measurements. A gas chromatography-flame ionization detector (GC-
FID, Agilent Technologies) provided hourly VOC measurements.
**3. Results and discussion**
**3.1. General $SF_6^-$ CIMS field performance**
**3.1.1. $SF_6^-$ ion chemistry with organic acids**

The underlying aspect of CIMS measurements of atmospheric constituents is the

use of ion-molecule reactions to selectively ionize compounds of interest in the complex
matrix of ambient air and produce characteristic ions. The reactions of $SF_6^-$ with the organic
acids (HX) gave similar products to those reported previously for $SF_6^-$ reactions with



inorganic acids (Huey et al., 1995): $SF_5^-$, $X^-$ and $X^-\bullet HF$ where $X^-$ is the conjugate base of
the organic acid (reactions 1a-c).
$\quad$ $SF_6^- + HX \rightarrow X^-\bullet HF + SF_5$ $\hspace{4cm}$ (1a)
$\quad$ $SF_6^- + HX \rightarrow X^- + HF + SF_5$ $\hspace{4cm}$ (1b)
$\quad$ $SF_6^- + HX \rightarrow SF_5^- + HF + X$ $\hspace{4cm}$ (1c)
The $SF_5^-$ ion (m/z 127, reaction 1c) is a common reaction product of the reactions of $SF_6^-$
with many species and is probably thermodynamically driven by the formation of HF
(Huey et al., 1995). Unfortunately, the production of $SF_5^-$ does not allow for the selective
detection of any atmospheric species. In addition, the larger the branching ratio of the $SF_5^-$
channel, the lower the CIMS sensitivity to an individual acid since the effective rate
constants for the $X^-$ and $X^-\bullet HF$ channels are lower.
$\quad$ The reaction of $SF_6^-$ with formic acid and oxalic acid also produced $SF_4^-$ ions (m/z
108). These reactions are probably thermodynamically driven by the formation of $CO_2$ and
HF:
$\quad$ $SF_6^- + HC(O)OH \rightarrow SF_4^- + CO_2 + 2HF$ $\hspace{3cm}$ (2)
$\quad$ $SF_6^- + HO(O)CC(O)OH \rightarrow SF_4^- + 2CO_2 + 2HF$ $\hspace{2cm}$ (3)
$\quad$ We used the $X^-$ and/or $X^-\bullet HF$ ions to determine ambient organic acid concentrations
since these ions are characteristic of the individual acids. For all the organic acids, the $X^-$
$\bullet HF$ ion signal is substantially lower than that of the $X^-$ ion for the conditions in this study.
However, this is probably largely due to the relatively high collision energy used in the
CDC which led to efficient dissociation of the fluoride adducts to form $X^-$ ions.
Consequently, only the proton transfer channel (1b) is used to quantify most of the organic
acids in the field study. The exceptions are formic and acetic acid as discussed in section
3.2.1 and 3.2.2 Table 1 shows a summary of the sensitivities of $X^-$ and $X^-\bullet HF$ ions of
organic acids relative to that of the $F_2{}^{34}SO_2^-$ ion.
**3.1.2. Characterization of interferences**





SF$_6^-$ is very sensitive to many trace atmospheric species but its reactions with water
vapor and O$_3$ when sampling ambient air can lead to issues both with selectivity and
stability. For example, SF$_6^-$ reacts nonlinearly with water vapor to form a series of F$^-$•(HF)$_n$
cluster ions (Huey et al., 1995; Arnold and Viggiano, 2001). SF$_6^-$ also reacts efficiently
with O$_3$ to form O$_3^-$, which is rapidly converted to CO$_3^-$ in ambient air (Slusher et al., 2001).
These reactions can deplete SF$_6^-$ as well as form a variety of potentially interfering ions
that depend on more abundant atmospheric species. For these reasons, efforts were made
to minimize interferences by limiting reaction times and the flow sampled into the CIMS.
This was accomplished by sampling only 0.3 L min$^{-1}$ of air through the variable orifice into
the flow tube which was maintained at a pressure of ~13 mbar. Figure 2 shows a mass
spectrum of ambient air. Interference peaks at m/z 39 (F$^-$•(HF) and CO$_3^-$, respectively) can
be attributed to the presence of water and O$_3$, respectively. Figure S2a shows the time series
of the $^{34}$SF$_6^-$ reagent ion signal and ambient water vapor concentration for the entire field
study. Despite fluctuations in ambient water vapor concentrations, the $^{34}$SF$_6^-$ reagent ion
signal was relatively constant for the entire field study with a standard deviation of < 3%.
This indicates that the reaction of SF$_6^-$ with ambient water vapor (and O$_3$) did not
significantly deplete the SF$_6^-$ reagent ions during the field study.
The F$_2{}^{34}$SO$_2^-$ ion signal was used to monitor the CIMS SO$_2$ sensitivity during the
field study. Figure S2b shows the time series of the F$_2{}^{34}$SO$_2^-$/$^{34}$SF$_6^-$ ion signal ratio obtained
in calibration measurements. There is a noticeable increase in the F$_2{}^{34}$SO$_2^-$/$^{34}$SF$_6^-$ ion signal
ratio on 28 Sept 2016, indicating an increase in the CIMS instrument sensitivity. The
increase in CIMS instrument sensitivity is due to the decrease in ambient water vapor
concentrations on 28 Sept 2016 (Fig. S2a). Previous laboratory and field studies showed
that this was due to the hydrolysis of F$_2{}^{34}$SO$_2^-$, which led to the loss of this ion and
diminished sensitivity at higher levels of ambient water vapor (Arnold and Viggiano, 2001;
Slusher et al., 2001). However, the SO$_2$ sensitivity at F$_2{}^{34}$SO$_2^-$ only varied within a factor
of two for the entire field study with a clear relationship to water vapor (Fig. S2c). The X$^-$
and X$^-$•HF ions of formic and acetic acids do not show any obvious dependence on ambient
water vapor concentration during calibration measurements. Therefore, we do not expect
the sensitivities of the X$^-$ and X$^-$•HF ions of the studied organic acids to depend on ambient
water vapor concentrations. We accounted for water vapor dependence of the F$_2{}^{34}$SO$_2^-$ ion



signal in our post-field calibrations where the response of the CIMS acid signals were
measured relative to the of the $^{34}SO_2$ sensitivity.

**3.2. Ambient measurements**


**3.2.1. Formic acid**


Figure 2 shows typical mass spectra obtained under background and measurement
modes during the field study. The $SF_6^-$ reagent ion is present at m/z 146. One of the
prominent species in the mass spectrum is formic acid, which is detected as $HCOO^-$ and
$HCOO^- \cdot HF$ at m/z 45 and 65, respectively. Our laboratory studies demonstrated that the
reaction of formic acid with $SF_6^-$ also produced a large fraction of $SF_4^-$ ions at m/z 108.
The reaction of $SF_6^-$ with oxalic acid also produced $SF_4^-$ ions, but its $SF_4^-$ product ion yield
is low and gas phase oxalic acid is not present in large concentrations. In addition, $SF_4^-$ is
present in the mass spectrum obtained under background mode but the $SF_4^-$ background
ion signals are lower than those typically observed in measurement mode at the Yorkville
site. As a result, we determined the ambient formic acid concentrations using the $HCOO^-$,
$HCOO^- \cdot HF$ and $SF_4^-$ ions. Figure 3a shows a scatter plot comparing the ambient formic
acid concentrations measured at Yorkville using the $HCOO^-$, $HCOO^- \cdot HF$ and $SF_4^-$ ions.
Linear regression analysis reveals that the formic acid concentrations determined by the
three ions are highly correlated ($R^2 = 0.99$) with slopes exhibiting a near 1:1 correlation.
The excellent correlation between these three ions and the agreement with laboratory data
indicates that formic acid is selectively measured by this method.
The time series of formic acid, temperature and solar radiation measured at
Yorkville are shown in Fig. 3b. Formic acid concentrations ranged from 0.04 to 4 ppb
during the field study, with strong and consistent diurnal trends. The day-to-day variability
in formic acid concentrations are associated with changes in solar radiation and
temperature. Higher formic acid concentrations are measured during warm and sunny days,
similar to formic acid measurements performed in Centreville, rural Alabama during the
2013 Southern Oxidant Aerosol Study (SOAS) (Brophy and Farmer, 2015; Millet et al.,
2015). Figure 3c shows the study-averaged diurnal profiles of formic acid and solar
irradiance. Formic acid started to increase at 7:30, which coincided with a sharp increase





in solar irradiance. Concentrations continued to increase throughout the day and peaked at
18:30, which coincided with the approximate time just before solar irradiance reached zero.
Formic acid then decreased continuously throughout the night.

The immediate early-morning increase in formic acid observed in this field study

is similar to that seen during the SOAS study (Millet et al., 2015). However, there are some
differences in the formic acid diurnal cycles measured in this field study and the SOAS
study. Formic acid peaked at 15:30 during SOAS, approximately 3 hours before solar
irradiance decreased to zero. In contrast, formic acid concentrations only started to
decrease at sunset (at 19:30) in this study. This suggests that there may be differences in
the types and/or magnitudes of formic acid sources and sinks in this two field studies. Land
cover and/or land use differences may have contributed to differences in formic acid
sources and sinks at the Centreville and Yorkville field sites. The area surrounding the
Yorkville field site is covered primarily by hardwood mixed with farmland and open
pastures. In contrast, the Centreville field site is surrounded by forests comprised of mixed
oak-hickory and loblolly trees (Hansen et al., 2003). It is also possible that seasonal
differences contributed to differences in formic acid sources and sinks in the two field
studies. The SOAS campaign took place in the middle of summer (1 June to 15 July 2013)
when biogenic emissions are typically higher while this field study took place in early fall
when biogenic emissions are lower due to cooler temperatures. For example, the average
concentration of isoprene (a formic acid source) in this study (1.21 ppb) is lower than that
in SOAS (1.92 ppb (Millet et al., 2015)). Despite these differences, our overall results are
similar to the formic acid measurements performed in SOAS in both magnitude and diurnal
variability.
**3.2.2. Acetic acid**

Acetic acid is detected with $SF_6^-$ as $CH_3COO^-$ and $CH_3COO^- \cdot HF$ at m/z 59 and 79,

respectively. However, these ions are subject to interferences from the reaction of $SF_6^-$ with
water vapor present in the sampled ambient air. Two of these interfering ions $F^- \cdot (HF)_2$ and
$F^- \cdot (HF)_3$ occur at m/z 59 and 79, respectively. As discussed earlier, we minimized the
impact of these interferences by diluting the sample flow into the CIMS and running the
CDC at a high collision energy to dissociate the HF cluster ions. As expected from cluster



bond strengths, we found that larger HF cluster ions dissociated more easily than smaller
ones. For example, at a CDC electric field of ~113 V cm$^{-1}$ (the configuration used in this
field study), virtually all of the F$^-$•(HF)$_3$ cluster ions dissociated while very few of the F$^-$
•(HF) cluster ions dissociated. This indicates that the m/z 79 channel for acetic acid is more
immune to interference from water vapor than the m/z 59 channel. This is supported by the
observation that the background ion signal at m/z 59 ($R^2 = 0.50$) is more highly correlated
with ambient water vapor concentrations than the background ion signal of m/z 79 ($R^2 =$
0.30). In addition, the m/z 59 ion is subjected to interference from the reaction of SF$_6^-$ with
$O_3$ present in the sampled ambient air. SF$_6^-$ reacts with $O_3$ in the presence of $CO_2$ to form
$CO_3^-$ at m/z 60 (Slusher et al., 2001). As shown in Fig. 2, the large $CO_3^-$ peak at m/z 60 is
a potential interference to the m/z 59 signal. As the background scrubber also removed $O_3$
from the ambient air, there is a large difference in the m/z 60 ion signal between the
measurement and background modes (~100,000 Hz). Thus, even a few percent bleed over
of m/z 60 to m/z 59 can lead to an over-estimation of ambient acetic acid concentrations.
For these reasons, we used m/z 79 (X$^-$•HF) to determine ambient acetic acid concentrations
even though this channel has a lower sensitivity than the m/z 59 channel (X$^-$).

The time series of acetic acid, temperature and solar radiation measured at

Yorkville are shown in Fig. 4a. Acetic acid concentrations ranged from 0.03 to 3 ppb during
the field study. The day-to-day variability in acetic acid concentrations resembled the
behavior of formic acid concentrations, with higher concentrations being measured during
warm and sunny days. Figure 4b shows the study-averaged diurnal profiles of acetic acid
and solar irradiance. The diurnal profile of acetic acid is similar to that of formic acid with
a more pronounced evening maximum. Acetic acid started to increase at 7:30 and built up
through the day, peaking at 19:30 and decreased continuously overnight. In general, acetic
acid concentrations are well correlated with ($R^2 = 0.67$) and comparable in magnitude (~60
% on average) to formic acid. The study-averaged formic acid/acetic acid concentration
ratio (1.65) is comparable to ratios from previous field studies in rural and urban
environments (Talbot et al., 1988; Talbot et al., 1995; Granby et al., 1997; Khare et al.,
1999; Talbot et al., 1999; Baboukas et al., 2000; Singh et al., 2000; Kuhn et al., 2002;
Baasandorj et al., 2015; Millet et al., 2015).



### 3.2.3. Larger organic acids

In addition to formic acid and acetic acid, eight other ions were monitored during the field study: m/z 73, 75, 87, 89, 101, 103, 117 and 131. These ions were chosen as they had significant signals when ambient air was sampled and were not obviously formed from $SF_6^-$ reaction with water vapor or $O_3$. Since the CIMS utilized in this study only had unit mass resolution, these ions are the sum of all organic acid isomers and isobaric organic acids of the same molecular weight as well as other product ions from species that might react with $SF_6^-$. However, real-time ion chromatography measurements of aerosol composition performed during the field study demonstrated the presence of particulate oxalic, malonic, succinic and glutaric acids (Nah et al., 2018). For this reason, for m/z 89, 103, 117 and 131 ions, we assigned them as $X^-$ ions of oxalic, malonic, succinic and glutaric acids, respectively. As these organic acids have low vapor pressures, their gas-phase concentrations are expected to be lower than their particle-phase concentrations, though their gas-particle ratios will depend on thermodynamic conditions (Nah et al., 2018). Particulate formic acid and acetic acid were also detected by ion chromatography during the field study, but were at much lower concentrations relative to the gas phase (Nah et al., 2018). For simplicity, we also denoted m/z 73, 75, 87 and 101 ions as $X^-$ ions of propionic, glycolic, butyric and valeric acids, respectively, for the remainder of this paper. These organic acids have previously been measured in rural and urban environments (Kawamura et al., 1985; Veres et al., 2011; Brophy and Farmer, 2015). However, we note that these assignments are speculative. Post-field calibration measurements were used to estimate the ambient concentrations of these organic acids.

Figure 5 shows the time series and diurnal profiles of oxalic, butyric, glycolic, propionic and valeric acids measured during the field study. Daytime concentrations of these organic acids ranged from a few tens of ppt to several hundred ppt. The time series of ion signals of malonic, succinic and glutaric acids are shown in Fig. S3. Concentrations of these organic acids are not available since calibrations were not performed for these compounds. The eight organic acids displayed very similar day-to-day variability as formic and acetic acids, with higher concentrations (or ion signals) being measured on warm and sunny days. The diurnal profiles of all the measured organic acids have similar diurnal



trends, with their concentrations reaching a maximum between 17:30 and 19:30 and rapidly
decreasing after sunset.
**3.2.4. Comparison with WSOC$_g$**

WSOC$_g$ measurements were performed during the field study using a MIST

chamber coupled to a TOC analyzer. The study average WSOC$_g$ was 3.6 ± 2.7 $\mu$gC m$^{-3}$,
slightly lower than that measured during the SOAS study (4.9 $\mu$gC m$^{-3}$) (Xu et al., 2017),
and approximately four times lower than that measured in urban Atlanta, Georgia (13.7
$\mu$gC m$^{-3}$) (Hennigan et al., 2009). Despite being comparable in magnitude, the diurnal
profiles of WSOC$_g$ measured in this study and the SOAS study are different. WSOC$_g$
measured in the SOAS study decreased at sunset, while WSOC$_g$ measured in this study
decreased 2 hours after sunset. Differences in WSOC$_g$ concentrations and diurnal profiles
at the three different sites may be due to differences in emission sources as a result of
different measurement periods, land use and/or land cover.

To estimate the fraction of WSOC$_g$ that is comprised of organic acids, the total

organic carbon contributed by formic, acetic, oxalic, butyric, glycolic, propionic and
valeric acids is compared to the WSOC$_g$ measurements. Figures 6a and 6b show the time
series and diurnal profiles of WSOC$_g$ and the organic carbon contributed by the measured
organic acids. Formic and acetic acids comprised majority of the total organic carbon
contributed by the measured organic acids (study averages of 31 and 38 %, respectively).
Assuming that the measured organic acids are completely water-soluble, the carbon mass
fraction of WSOC$_g$ comprised of these organic acids ranged from 7 to 100 %. Based on the
orthogonal distance regression slope shown in Fig. 6c, the study-averaged carbon mass
fraction of WSOC$_g$ comprised of the measured organic acids is 30 %. The total organic
carbon contributed by the measured organic acids are moderately correlated with WSOC$_g$
($R^2$ = 0.38). This is likely due to the presence of other water-soluble gas phase species
(with different day-to-day variability from the organic acids) that contribute to the WSOC$_g$.
This is supported by slight differences in the diurnal profiles of WSOC$_g$ and the organic
carbon contributed by the organic acids (Fig. 6b). While the diurnal profiles of WSOC$_g$
and the organic carbon contributed by the organic acids have similar general shapes,
WSOC$_g$ peaked at 21:30, approximately 2 hours after the solar irradiance have decreased



to zero. In contrast, the organic carbon contributed by the organic acids start to decrease at
sunset (at 19:30).

**3.2.5. SO₂ and HNO₃ observations**

**3.2.5. SO$_2$ and HNO$_3$ observations**

In addition to evaluating the field performance of the SF$_6^-$ CIMS technique in gas-

phase organic acid measurements, another focus of this study was to investigate the
possible sources of the measured organic acids. For this reason, HNO$_3$ and SO$_2$ (two
common anthropogenic tracers) were also measured by SF$_6^-$ CIMS during the field study.
Correlations between the concentrations of organic acids and those of HNO$_3$ and SO$_2$ were
then examined to determine if the organic acids were anthropogenic in nature (section 3.3).
While their reactions with SF$_6^-$ have multiple product channels (Huey et al., 1995), only
the NO$_3^-$•HF (m/z 82) and F$_2$SO$_2^-$ (m/z 102) ions were used for quantitative purposes:

$SF_6^- + HNO_3 \rightarrow NO_3^- \bullet HF + SF_5$                    (4)

$SF_6^- + SO_2 \rightarrow F_2SO_2^- + SF_4$                    (5)

Figure S4 shows the time series of SO$_2$ and HNO$_3$ measured during the field study.

As expected at a rural site, SO$_2$ and HNO$_3$ concentrations are low most of the time (study
averages of 0.23 and 0.18 ppb, respectively). However, there were occasional periods when
the site was impacted by anthropogenic pollution. In particular, there are spikes in both
SO$_2$ and HNO$_3$ concentrations throughout the study that corresponded to the site being
impacted by power plant or urban emissions. Outside of these anthropogenic spikes, HNO$_3$
showed a clear diurnal profile with a maximum at approximately 12:30, consistent with
local photochemical production.

**3.3. Potential sources of organic acids**

**3.3. Potential sources of organic acids**

Correlation analysis on organic acid concentrations can provide insights on their

sources. Figure 7 shows that the concentration of formic acid is strongly correlated with
those of the other measured organic acids (R$^2$ = 0.68 to 0.89). This suggests that these
organic acids have the same or similar sources at Yorkville. The sources of organic acids
can be biogenic or anthropogenic in nature. To determine if the primary sources of organic
acids are of biogenic or anthropogenic origin, we first examined the correlations of organic



acid concentrations with those of anthropogenic pollutants CO, $SO_2$, $O_3$ and $HNO_3$. CO
and $SO_2$ are common tracers for combustion sources. The organic acid concentrations are
poorly correlated with CO (Fig. S5, $R^2 = 0.03$ to 0.15) and $SO_2$ (Fig. S6, $R^2 = 0.01$ to 0.24),
indicating that primary emissions from combustion are a minor source of organic acids in
Yorkville. $HNO_3$ and $O_3$ are common photochemical tracers of urban air masses. The
organic acid concentrations are weakly correlated with $O_3$ (Fig. S7, $R^2 = 0.11$ to 0.32) and
$HNO_3$ (Fig. S8, $R^2 = 0.33$ to 0.56). In addition, there is no noticeable increase in organic
acid concentrations during periods of elevated CO, $SO_2$, $O_3$ and $HNO_3$ concentrations when
the site was impacted by pollution plumes. Together, these results indicate that the primary
sources of organic acids in Yorkville are likely not anthropogenic in nature.
Diurnal profiles of the measured organic acids suggest that their sources are linked
to higher daytime temperatures and/or photochemical processes. Figure 8 compares the
concentrations of organic acids against ambient temperatures measured during the study.
Since there was a noticeable decrease in mean ambient temperatures starting on 28 Sept
2016, we grouped the datasets into two time periods (3 to 27 Sept and 28 Sept to 12 Oct)
to better evaluate the effect of temperature on organic acid concentrations. The average
temperature in the first time period (3 to 27 Sept) is 24.8 °C (32.6 °C max, 18.1 °C min),
while the average temperature in the second time period (28 Sept to 12 Oct) is 19.5 °C
(28.4 °C max, 9.5 °C min). We find that organic acid concentrations are on average higher
and more highly correlated with temperatures in the warmer first time period ($R^2 = 0.40$ to
0.63) compared to the cooler second time period ($R^2 = 0.18$ to 0.59). These observations
can be explained by temperature-dependent emissions of organic acids and their BVOC
precursors. Previous studies have shown that emissions of organic acids and their BVOC
precursors depend strongly on light and temperature, with substantially lower
concentrations being emitted in the dark and/or at low temperatures (Kesselmeier et al.,
1997; Kesselmeier, 2001; Sindelarova et al., 2014). We find that the concentration of
isoprene, which was the dominant BVOC in Yorkville, has a somewhat similar diurnal
profile as the organic acids and decreased with temperature on 28 Sept 2016 (Fig. S9). In
addition, the concentrations of formic and acetic acids are moderately correlated with that
of isoprene ($R^2 = 0.42$ and 0.40, respectively) (Fig. S10).



Multiphase photochemical aging of ambient organic aerosols can also be a source

of gas-phase organic acids (Eliason et al., 2003; Ervens et al., 2004; Molina et al., 2004;
Lim et al., 2005; Park et al., 2006; Walser et al., 2007; Sorooshian et al., 2007; Vlasenko
et al., 2008; Pan et al., 2009; Sorooshian et al., 2010). Organic acids may be formed in the
particle phase during organic aerosol photochemical aging, with subsequent volatilization
into the gas phase. Real-time ion chromatography measurements of aerosol composition
demonstrated the presence of particulate formic, acetic, oxalic, malonic, succinic and
glutaric acids (Nah et al., 2018). However, since the ratios of gas-phase formic and acetic
acid mass concentration to the total organic aerosol mass concentration are large (study
averages of 40 and 35 %, respectively) (Nah et al., 2018), it is unlikely that organic aerosol
photochemical aging is a large source of formic and acetic acids. In contrast, the ratios of
gas-phase oxalic, malonic, succinic and glutaric acids mass concentration to the total
organic aerosol mass concentration are small, suggesting that organic aerosol
photochemical aging may be an important source of these gas-phase organic acids.

In summary, the temperature dependence and diurnal profile of organic acid

concentrations combined with poor correlations between organic acid concentrations and
those of anthropogenic pollutants CO, $SO_2$, $O_3$ and $HNO_3$ strongly suggest that the primary
sources of gas-phase organic acids at Yorkville are biogenic in nature. However, our data
alone does not allow us to determine if the organic acids are a result of direct emissions or
photochemical oxidation of other BVOC emissions and/or organic aerosols. Partitioning
of these organic acids between the gas and particle phases will be discussed in a
forthcoming paper (Nah et al., 2018).
**4. Summary**

$SF_6^-$ reacted with all of the studied organic acids to produce product ions that were

characteristic of the individual acids (i.e., $X^-$ or $X^-\cdot HF$). These reactions all occurred at less
than the maximum collisional rate due to significant yields of $SF_5^-$ and $SF_4^-$, which reduced
the sensitivity of the method. For the conditions employed in this study, the sensitivities of
$X^-$ and $X^-\cdot HF$ ions of the organic acids relative to that of the $F_2^{34}SO_2^-$ ion (study-averaged
sensitivity $2928 \pm 669$ Hz ppb$^{-1}$) ranged from 0.04 to 2.18. The detection limits of the
organic acids were approximated from 3 times the standard deviation values ($3\sigma$) of the ion
signals obtained during background measurements. Reasonable limits of detection for 2.5
min integration periods (1 to 60 ppt) were obtained for all the organic acids studied. Water
vapor and $O_3$ can lead to interferences with this method but for the conditions employed in
this study, they were largely limited to acetic acid measurements at m/z 59. However,
fluctuations in ambient water vapor can also lead to changes in sensitivity for the detection
of some species (e.g., $SO_2$). Uncertainties in organic acid concentrations originate primarily
from calibration measurements and ranged from 9 to 22 %. Overall, the tractable mass
spectra obtained by the $SF_6^-$ CIMS method coupled with reasonable limits of detection and
the high correlations observed between the individual organic acids demonstrated the
potential of this method. Obvious next steps for the $SF_6^-$ CIMS method are to compare it
to other measurement methods for organic acids and to deploy the $SF_6^-$ ion chemistry to a
higher resolution time-of-flight mass spectrometer to reduce the potential for interferences.
The $SF_6^-$ CIMS method was deployed for measurements of gas phase organic acids
in a mixed forest-agricultural area in Yorkville, Georgia from Sept to Oct 2016. The
organic acids measured in the field study were formic and acetic acids. In addition,
measurements tentatively assigned to oxalic, butyric, glycolic, propionic, valeric, malonic,
succinic and glutaric acids were performed. Ambient concentrations of these organic acids
ranged from a few ppt to several ppb. All the organic acids exhibited similar strong diurnal
trends. Organic acid concentrations built up throughout the day, peaked between 17:30 and
19:30 before decreasing continuously overnight. Strong correlations between organic acid
concentrations indicated that these organic acids likely have the same or similar sources at
Yorkville. We concluded that the organic acids were likely not due to anthropogenic
emissions since they were poorly correlated with anthropogenic pollutants and their
concentrations were not elevated when the site was impacted by pollution plumes. Higher
organic acid concentrations were measured during warm and sunny days. Organic acid
concentrations were strongly correlated with temperature during the first month of the
study when ambient temperatures were high. Together, our results suggested that the
primary sources of organic acids at Yorkville were biogenic in nature. Direct biogenic
emissions of organic acids and/or their BVOC precursors were likely enhanced at high
ambient temperatures, resulting in the observed variability of organic acid concentrations.
Another potential source is the production of organic acids in the particle phase from the





multiphase photochemical aging of organic aerosols followed by evaporation to the gas
phase, though this source is likely not a large source of formic and acetic acids. However,
given the inability of current models and photochemical mechanisms to explain formic acid
observations in the Southeastern U.S. (Millet et al., 2015), it is unlikely that our
observations of formic acid and larger organic acids can be explained as well. Further work
(i.e., laboratory, field and modeling studies) is needed to determine how organic acids are
formed in the atmosphere.
**5. Acknowledgements**

The authors thank Eric Edgerton (Atmospheric Research and Analysis, Inc.) for

providing CO, $O_3$ and VOC measurements and meteorological data.
**6. Funding**

This publication was developed under US Environmental Protection Agency (EPA)

STAR Grant R835882 awarded to Georgia Institute of Technology. It has not been
formally reviewed by the EPA. The views expressed in this document are solely those of
the authors and do not necessarily reflect those of the EPA. EPA does not endorse any
products or commercial services mentioned in this publication.
**7. Competing financial interests**

The authors declare no competing financial interests.

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





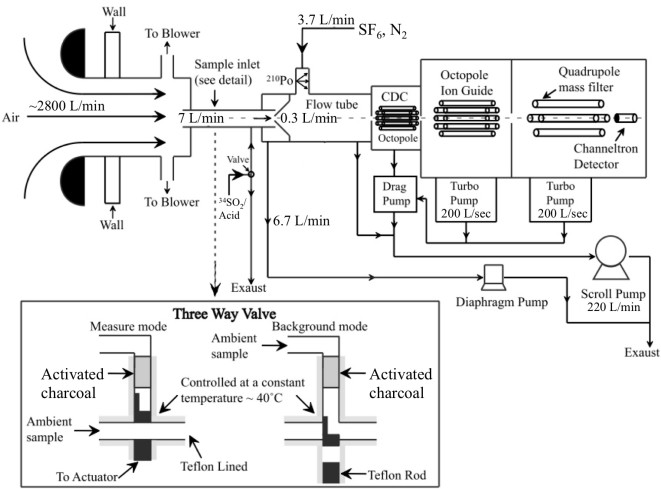


**Figure 1:** The CIMS instrument and inlet configuration used in the field study. The automated three-way sampling valve is shown in the inset. The figure was adapted from Liao et al. (2011).












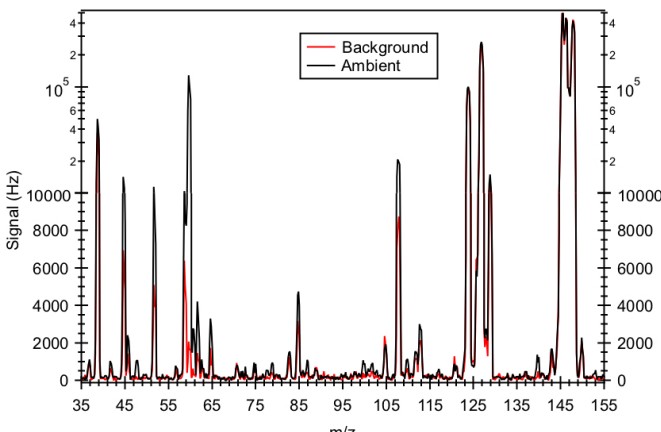


**Figure 2:** Mass spectrum of ambient air and background measured in Yorkville, Georgia
on 8 Sept 2016 using $SF_6^-$.

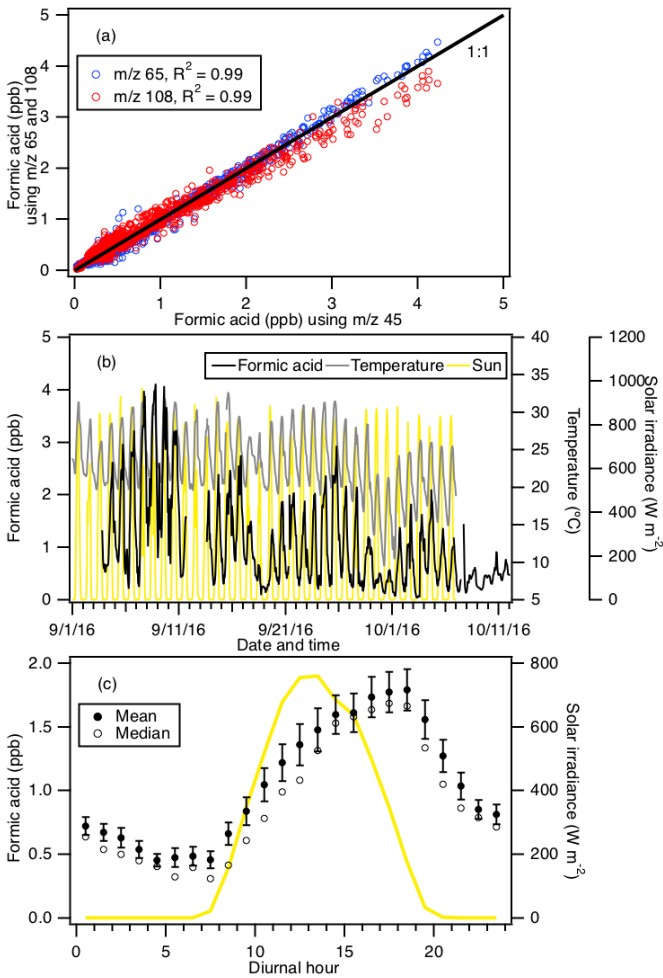


**Figure 3:** (a) Scatter plot comparison of ambient formic acid concentrations determined using mass peaks m/z 45, 65 and 108. The three datasets correlated well with one another ($R^2 = 0.99$). Linear regression of the data gave slopes of 1 (for m/z 65) and 0.95 (for m/z 108), indicating that all three mass peaks can be used to determine the formic acid concentration. (b) Time series of formic acid concentration, temperature and solar irradiance. All the data are displayed as 1-hour averages. (c) Diurnal profiles of formic acid concentration (symbols) and solar irradiance (yellow line). All the concentrations represent averages in 1-hour intervals and the standard errors are plotted as error bars.





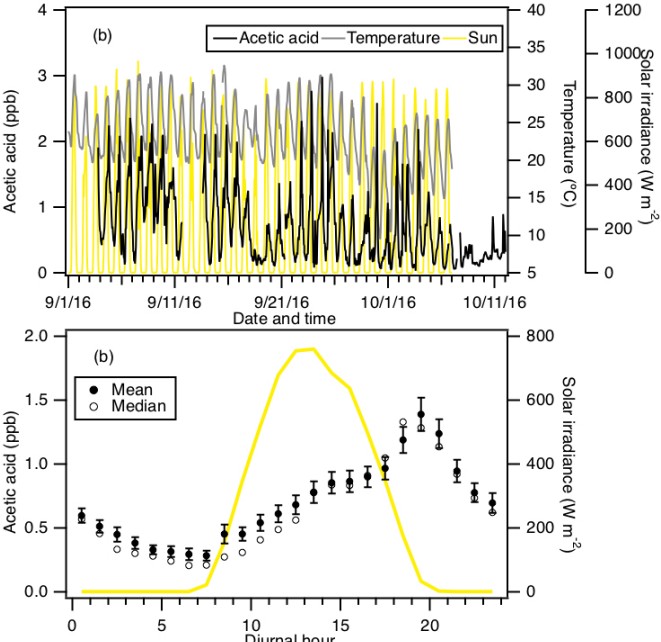

903

**Figure 4:** (a) Time series of acetic acid concentration, temperature and solar irradiance. All the data are displayed as 1-hour averages. (c) Diurnal profiles of acetic acid (symbols) and solar irradiance (yellow line). All the concentrations represent averages in 1-hour intervals and the standard errors are plotted as error bars.



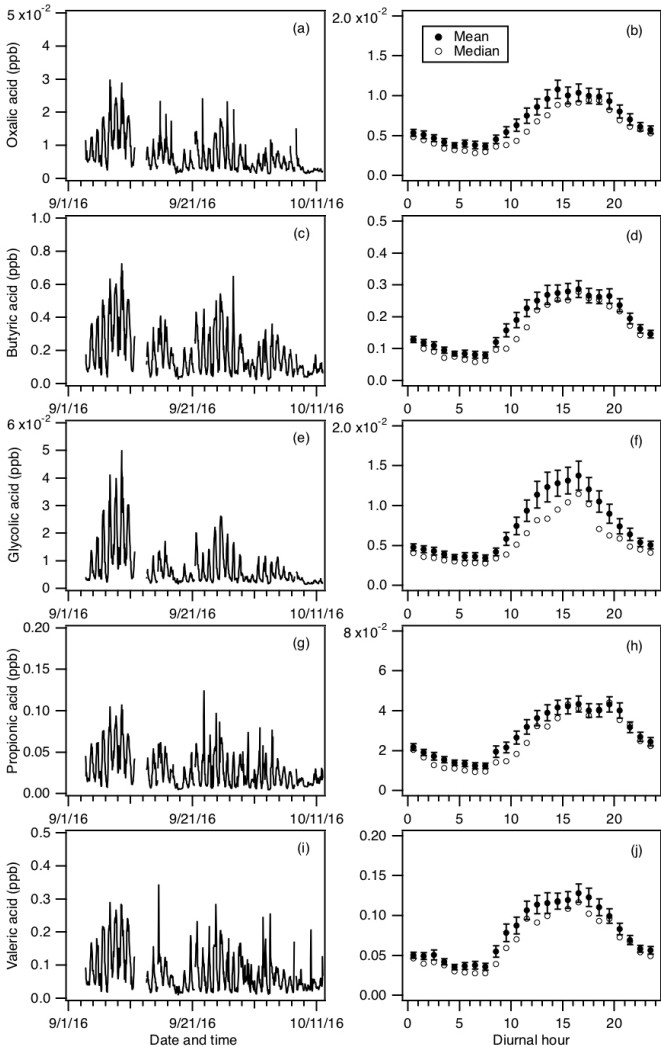

908

**Figure 5:** Time series of concentrations of (a) oxalic, (c) butyric, (e) glycolic, (g) propionic, and (i) valeric acids measured during the field study. All the data are displayed as 1-hour averages. Their corresponding diurnal profiles are shown in (b), (d), (f), (h) and (j), respectively. The diurnal profile concentrations represent averages in 1-hour intervals and the standard errors are plotted as error bars.



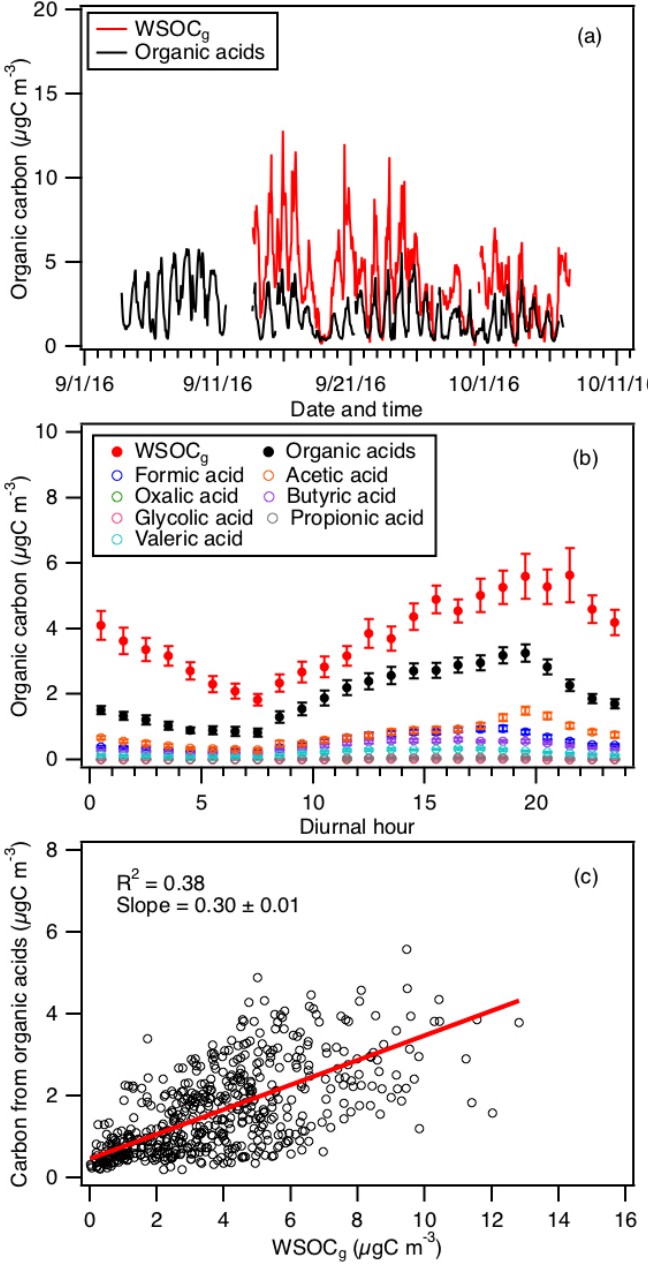

914

**Figure 6:** (a) Time series of $WSOC_g$ and the total organic carbon contributed by the

measured organic acids (i.e., formic, acetic, oxalic, butyric, glycolic, propionic and valeric

acids). All the data are displayed as 1-hour averages. (b) Diurnal profiles of $WSOC_g$ and

the total organic carbon contributed by the measured organic acids. Also shown are the



diurnal profiles of the organic carbon contributed by the individual measured organic acids.
All the concentrations represent the mean hourly averages and the standard errors are
plotted as error bars. (c) Scatter plot of total organic carbon contributed by the measured
organic acids with $WSOC_g$.

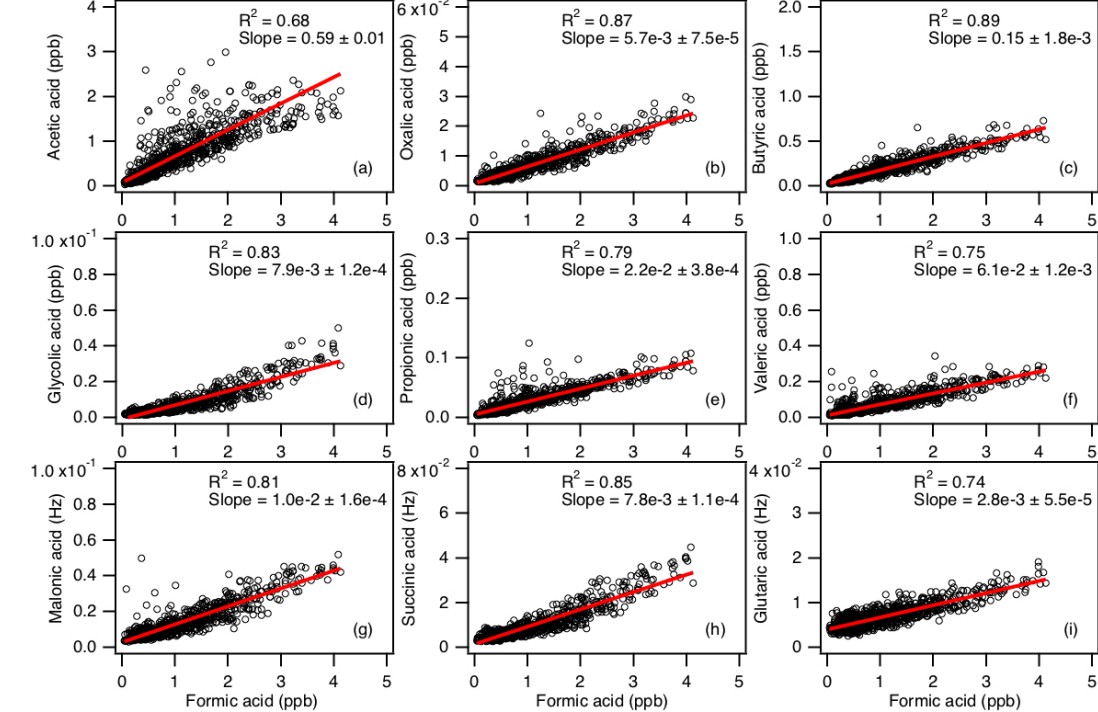

**Figure 7:** Scatter plots of concentrations (or signals) of (a) acetic, (b) oxalic, (c) butyric,
(d) glycolic, (e) propionic, (f) valeric, (g) malonic, (h) succinic, and (i) glutaric acids with
formic acid concentration. All the data are displayed as 1-hour averages. The data for
malonic, succinic and glutaric acids are presented as Hz normalized by the instrument's
sensitivity to $F_2{}^{34}SO_2$ since these organic acids were not calibrated. Red lines shown are
linear fits to the data.





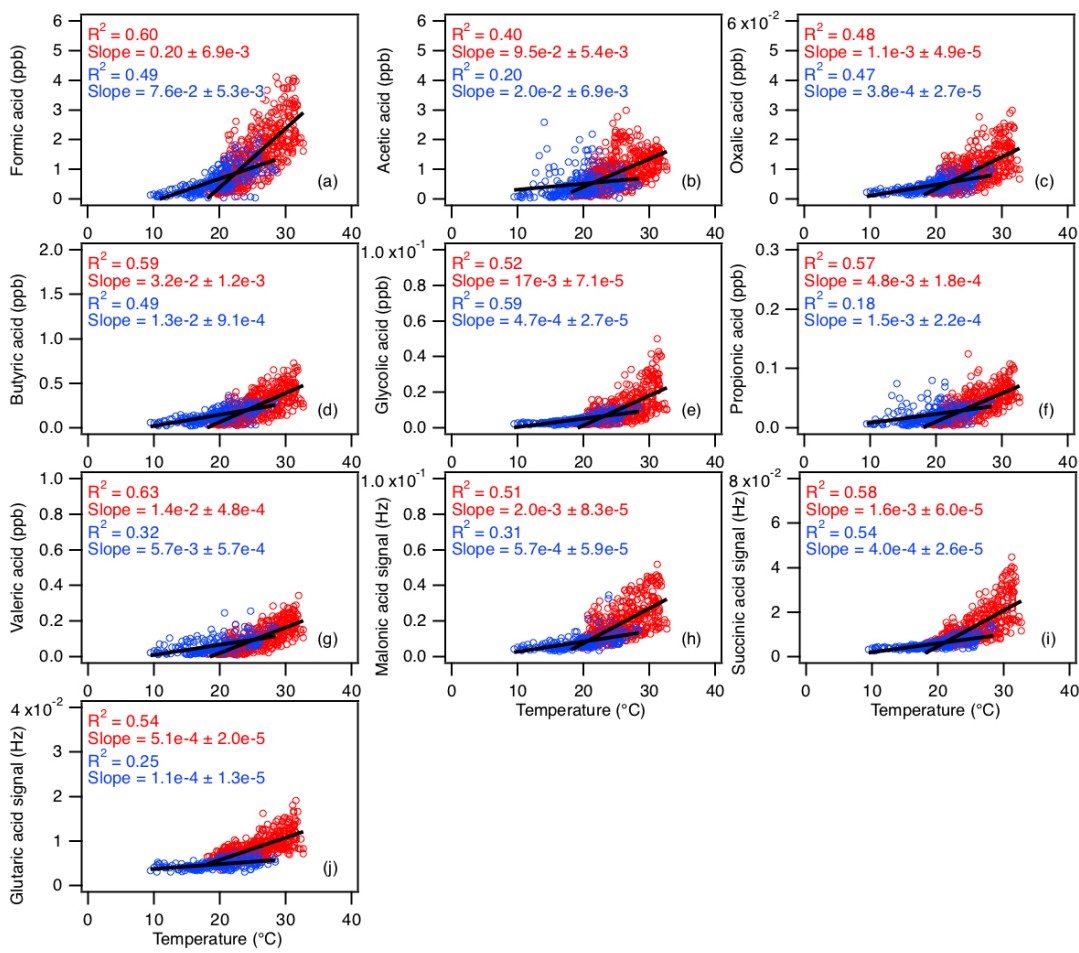

930

**Figure 8:** Scatter plots of concentrations (or signals) of (a) formic, (b) acetic, (c) oxalic, (d) butyric, (e) glycolic, (f) propionic, (g) valeric, (h) malonic, (i) succinic, and (j) glutaric acids with ambient temperature. The red symbols are data collected from 3 to 27 Sept, while the blue symbols are data collected from 28 Sept onwards. All the data are displayed as 1-hour averages. The data for malonic, succinic and glutaric acids are presented as Hz normalized by the instrument's sensitivity to $F_2^{34}SO_2$ since these organic acids were not calibrated. Black lines shown are linear fits to the datasets.







**Table 1:** Summary of organic acids of interest, their detection limits and sensitivities of
their $X^-$ and $X^-{\cdot}HF$ ions relative to the $F_2{}^{34}SO_2{}^-$ ion[a]

| Organic Acid | Detection limit (ppb)[b] | Sensitivity relative to the $F_2{}^{34}SO_2{}^-$ ion[c] | |
|---|---|---|---|
| | | $X^-$ | $X^-{\cdot}HF$ |
| Formic acid | 0.03 | 0.44 | 0.10 |
| Acetic acid | 0.06 | 0.50 | 0.10 |
| Oxalic acid | $1 \times 10^{-3}$ | 2.18 | 0.33 |
| Butyric acid | 0.03 | 0.14 | 0.04 |
| Glycolic acid | $2 \times 10^{-3}$ | 1.89 | 0.56 |
| Propionic acid | $6 \times 10^{-3}$ | 0.70 | 0.43 |
| Valeric acid | 0.01 | 0.26 | 0.12 |

[a]Only organic acids with calibration measurements are shown.
[b]Detection limits are approximated from 3 times the standard deviation values ($3\sigma$) of the
ion signals measured during background mode. Shown here are the average detection limits
of the organic acids for 2.5 min integration periods which corresponds to the length of a
background measurement at a 0.04 s duty cycle for each mass.
[c]Study-averaged sensitivity of the $F_2{}^{34}SO_2{}^-$ ion = 2928 ± 669 Hz ppb$^{-1}$