# Peer review of "Real-time measurements of gas-phase organic acids using SF₆⁻ chemical ionization mass spectrometry"

_Atmospheric Measurement Techniques, 2018_

## Referee Comment (RC1) · Anonymous Referee #1 · 15 Mar 2018

The manuscript "Real-time measurements of gas-phase organic acids using SF6-chemical ionization mass spectrometry" by Nah et al. presents a new measurement technique for organic acids and shows results from a field study. Demonstrating that SF6- is useful for measurements in a humid environment is an important new finding that may expand the capabilities of CIMS instruments for quantifying atmospheric trace gases, since previous reports had shown this reagent ion to be effective only in very dry locations with moderate ozone levels (like the Arctic). The subject matter is appropriate for AMT if the recommendations below are addressed, and the writing is easy to understand. Many experimental details and clarifications are needed to explain and justify this new technique. The discussion of the contribution of these organic

acid contributions to WSOC and the many figures that detail organic acid mixing ratios should be pared down, since the species identification here are speculative, and there may be many interferences that have not been adequately considered. Specific recommendations are detailed below.

Previous reports have noted that SF6- is a viable reagent ion only for dewpoints below -20 C. The flow tube is at lower pressure here, and that is said to minimize interferences with water vapor (line 167). This needs to be explained much more thoroughly. The very high dilution with dry nitrogen likely does much more to reduce the water vapor interferences, but this is never discussed. This high dilution must reduce sensitivity. Please discuss the trade-off between reduced sensitivity and increased selectivity. The flows as shown don't make sense, and volume flow and mass flow don't appear to be correctly distinguished. Accurately describing and discussing the flows is critical, since it is the altered gas flows that have made SF6- viable in a humid environment. Line 154 gives the sampled air flow as a volume flow, but the orifice is said to maintain a constant mass flow. A 220 l/min scroll pump is used to pump a very small flow of 4 L/min through the flow tube, as shown in Figure 1. I suspect that the N2 flow should be written as 3.7 slpm, rather than L/min. But even that doesn't make sense, since it would be ~300 L/min volume flow, and greater than the pump speed. The instrument is said to be similar to Liao et al 2011, but the flows are very different, and these should be discussed completely. Does the high dilution compromise time response, since there is a very small flow of ambient air into the flow tube? Does the large flow of N2 through the ion source increase reagent ion signal?

Many fundamental measurement characteristics are not discussed. Please state and demonstrate time response, and preferably show a calibration and a zero. Detection limits are given for 2.5 min. I don't understand the relation between the 2.5 min background and the 0.04 s duty cycle for each mass (line 946). Please describe the measurement frequency - how often is a measurement made? I'm confused as to whether these are 0.04 s measurements, 2.5 min measurements, or 1 hr measurements (as

used in the figures). And please be specific about the integration time for the mixing ratios. For example, are the range of mixing ratios in the abstract (line 22) for 1 hr averages? Detection limits and abundances in the figures are written in ppb, but for most cases, ppt would be a more appropriate unit and much easier to read.

Giving sensitivities relative to 34SO2 is confusing and seems unnecessary, especially since that is a different ion-molecule reaction than all the organic acid reactions studied here and it has a water dependence. Providing absolute sensitivities will be much more useful for anyone who wants to compare this ion chemistry with others. And please discuss how these sensitivities compare to other techniques.

Please discuss the many possible interferences to the organic acids listed. The discussion of calibrations for many compounds was valuable, but then compounds for which calibrations weren't successful (pyruvic, for example) were dismissed. It appears arbitrary to ignore compounds if the calibration didn't work. Those other compounds almost certainly contribute to the signals and make the calculations of WSOC fraction extremely unreliable. For example, the signals at oxalic likely are dominated by or include contributions from lactic acid (both mass 90), butyric includes contributions from pyruvic (both mass 88), propionic includes contributions from glyoxylic acid (both mass 74). The authors correctly note that the mass assignments are speculative. It is OK to speculate, but from then on it would be appropriate to report mixing ratios only as upper limits for that compound and then forego calculations that rely on quantifying abundance. The sensitivities of the reported compounds span a wide range (over an order of magnitude). Thus, if the signal at one of the reported masses comes instead from an isomer with a much higher or lower sensitivity, the organic acid abundance would be drastically over or under estimated. There needs to be considerably more work to justify the WSOC contributions. I don't understand the count rates and normalization shown in figures 6-8. How do the count rates get to be small fractions of 1 Hz? Please use a unit that is more accurate and describe how the values are determined.

I don't understand the reagent ion signal and the use of sulfur isotopes. 34SF6- is

said to be the reagent ion (line 338). Is isopically labeled SF6 used? Why do the magnitudes of the peaks at mass 145 and 147 appear to be similar in Figure 2? Please clearly state the magnitude of the reagent ion signal. From figure 2, it appears that the reagent ion signal is approx 400 kHz, and similar in magnitude to SF5-, CO3-. Wouldn't ion chemistry from those other ions also contribute to the reactions? A more thorough description of the reagent ion is necessary to understand how the ion chemistry can be dominated by SF6-. Figure 2 makes it appear that other ions could be substantial contributors to the ion chemistry.

Since the major measurement technique advance here is using SF6 in high water and ozone environments, there should be greater discussion of water and ozone mixing ratios in the main text, and the necessary modifications to make the ion chemistry work in this environment. Although the supplement shows some ozone and water data, simpler discussion of the range of ozone and water mixing ratios and the effect on the ion chemistry should be included in the manuscript. Please show how formic or acetic sensitivity varies with ozone and water - this will make a much more convincing case for a valuable technique than showing F2SO2 versus water in the supplement. One overlooked feature in the results is the differing diurnal variations of acetic and formic acids in figures 3-4. Rather than focus on organic acid budgets that rely on speculative assignments and suffer from unexamined interferences, a study of the diurnal variability of acetic and formic acids could prove interesting.

---

## Referee Comment (RC2) · Anonymous Referee #2 · 16 Mar 2018

Nah et al. describe the extension of SF6- ion chemistry for the selective detection of organic acids. The paper describes both laboratory and field characterization of the technique and highlights challenges specific to the ion chemistry that need to be accounted for when making ambient measurements. The paper is well suited for publication in AMT following the author's response to the comments raised here.

General Comments:

The authors make a strong statement (line 95) that new techniques for the real-time measurement of gas-phase organic acids are need due to deficiencies in existing CIMS based ion chemistry (acetate and iodide CIMS). The authors cite issues with acetate

[Figure]

CIMS in the detection of acetic acid and the wide range of sensitivities to different organic acids in iodide CIMS. After reading this paper it was not clear to me that SF6-has an advantage over these techniques. It was shown that interferences due to O3 hinder detection of acetic acid and there is an order of magnitude spread in sensitivity to various organic acids in Table 1. I think the authors need to better articulate how this technique is an advance over existing ion chemistry or acknowledge that it is a parallel approach to existing ion chemistry.

Given the focus of the journal, I think more emphasis on the experimental configuration of the CIMS should be given. Direct comparisons to existing measurements are always welcome, but the focus should remain on describing the details of the ion chemistry and or instrument operation.

Specific and technical comments:

Line 46: I would cite Molina et al. (2004), or Vlasenko et al. (2008) for the heterogeneous source of organic acids from the chemical aging of organic aerosol.

Line 166: Please elaborate on how the lower pressure (13 mbar) minimizes interferences in reactions of SF6- with water vapor.

Line 171: Rather than reporting voltages, it would be better to report electric fields or relative electric fields (E/N).

Line 182: Why was the background measurement period so long ($\sim$4 minutes). It would be helpful to show one of these in time. I would have expected that the background measurement period could be significantly shorter and still capture the baseline, unless there are inlet equilibration issues.

Line 183: What was the 1.12 ppm SO2 standard diluted to? Presumably calibrations were not done at this mixing ratio.

Line 184: Again, it would be more helpful to present as the concentration of FA or AA that is delivered instead of the permeation tube emission rates.

Line 308: For reactions 1a-c, should one think of these as separate reaction channels governed by ion-molecule kinetics or does every reaction proceed through 1a and the electric field strength of the CDC sets the ratio of the observed products. This may lead to strong deviations in the observed products based on instrument configuration.

Line 334: Can the mass (or molar) dilution constant be reported here instead of the inlet flow? This would help the reader understand how much ambient $O_3$ and $H_2O$ are reduced by the sampling geometry. Also, perhaps it would be helpful to more explicitly state how a reduction in ion-molecule reaction time is helpful. This would not help in sensitivity (assuming all reactions are at the collision limit), but presumably would help minimize secondary ion chemistry, correct?

Line 335: What is the uncertainty in the IMR and CDC pressure? Are these pressures also controlled?

References:

Molina, M. J., Ivanov, A. V., Trakhtenberg, S., and Molina, L. T.: Atmospheric evolution of organic aerosol, Geophys Res Lett, 31, 2004.

Vlasenko, A., George, I. J., and Abbatt, J. P. D.: Formation of volatile organic compounds in the heterogeneous oxidation of condensed-phase organic films by gas-phase OH, J Phys Chem A, 112, 1552-1560, 2008.

---

## Author Comment (AC1) · 24 May 2018

We greatly value the careful reading and the detailed comments provided by the referees. The responses to the comments of the referees in our direct reply (shown below) and within the revised manuscript (see marked copy) are provided below. The pages and lines indicated below correspond to those in the marked copy.

**Response to Referee 1 (Referees' comments are italicized)**

1. Referee comment: "*Many experimental details and clarifications are needed to explain and justify this new technique. The discussion of the contribution of these organic acid contributions to WSOC and the many figures that detail organic acid mixing ratios should be pared down, since the species identification here are speculative, and there may be many interferences that have not been adequately considered.*"

**Author response:** We agree that the assignment of some of the organic acids, with the exception of formic and acetic acids, are speculative. As such, we have revised the manuscript to be more circumspect about the organic acid contribution to $WSOC_g$. Please refer to our response to comment 7.

We have also added more experimental details and discussions on interferences in the revised manuscript. Please refer to our responses to comments on experimental details and interferences below.

2. Referee comment: "*Previous reports have noted that SF6- is a viable reagent ion only for dewpoints below -20 C. The flow tube is at lower pressure here, and that is said to minimize interferences with water vapor (line 167). This needs to be explained much more thoroughly. The very high dilution with dry nitrogen likely does much more to reduce the water vapor interferences, but this is never discussed. This high dilution must reduce sensitivity. Please discuss the trade-off between reduced sensitivity and increased selectivity. Does the high dilution compromise time response, since there is a very small flow of ambient air into the flow tube? Does the large flow of N2 through the ion source increase reagent ion signal?*"

**Author response:** The referee is correct in stating that there are tradeoffs between reduced sensitivity and increased selectivity when the sampled air is diluted by a large volume of $N_2/SF_6$ in the CIMS flow tube. The high flow (3.7 slpm) of $N_2/SF_6$ passed through the ion source increased the $SF_6^-$ reagent ion signal. This does increase sensitivity but the $N_2$ flow also dilutes the sampled air flow in the flow tube which reduces the sensitivity. We chose these conditions to bring the water interferences to an acceptable level. We don't see significant differences in the time response for sample air flow flows $\geq 2$ L min$^{-1}$ introduced into the flow tube because a large flow (7 L min$^{-1}$) is maintained in the sample inlet right up to the CIMS flow tube (Fig. 1). As requested, we have added more discussion on the tradeoffs between reduced sensitivity and increased selectivity when the sampled air is diluted by a large volume of $N_2$ in the CIMS flow tube and how the high dilution impacts time response in the revised manuscript:

**Page 13 line 389: "The 0.3 L min$^{-1}$ sampled air flow is diluted by 3.7 slpm of $N_2/SF_6$ flow in the flow tube. The ratio of the sampled air flow to the $N_2/SF_6$ flow introduced into the flow tube is approximately 1:13. While the high $N_2/SF_6$ flow (3.7 slpm) passed through the radioactive source into the flow tube increased the $SF_6^-$ reagent ion signal, the high dilution**

**of the sampled air flow in the flow tube reduced the CIMS instrument sensitivity by decreasing the number density of the analytes."**

3. Referee comment: "*The flows as shown don't make sense, and volume flow and mass flow don't appear to be correctly distinguished. Accurately describing and discussing the flows is critical, since it is the altered gas flows that have made SF6- viable in a humid environment. Line 154 gives the sampled air flow as a volume flow, but the orifice is said to maintain a constant mass flow. A 220 l/min scroll pump is used to pump a very small flow of 4 L/min through the flow tube, as shown in Figure 1. I suspect that the N2 flow should be written as 3.7 slpm, rather than L/min. But even that doesn't make sense, since it would be ~300 L/min volume flow, and greater than the pump speed. The instrument is said to be similar to Liao et al 2011, but the flows are very different, and these should be discussed completely.*"

**Author response:** The referee is correct in pointing out that our previously stated flows and pump rates were incorrect. A 367 L min$^{-1}$ scroll pump (Edward nXDS 20i) was used. The N$_2$/SF$_6$ flow was 3.7 slpm. These corrections have been made to the revised manuscript:

**Page 6 line 166: "The CIMS instrument was comprised of a series of differentially pumped regions: a flow tube, a collisional dissociation chamber, an octopole ion guide, a quadrupole mass filter and an ion detector. These sections were evacuated by a scroll pump (Edward nXDS 20i), a drag pump (Adixen MDP 5011) and two turbo pumps (Varian Turbo-V301), respectively. Ambient air was drawn continuously into the flow tube. A flow of 3.7 slpm of N$_2$ containing a few ppm of SF$_6$ (Scott-Marrin Inc.) was passed through a $^{210}$Po ion source into the flow tube. SF$_6^-$ anions, which were produced via associative electron attachment in the $^{210}$Po ion source, reacted with the sampled ambient air in the flow tube to generate analyte ions."**

[Figure]

**Figure 1: The CIMS instrument and inlet configuration used in the field study. The automated three-way sampling valve is shown in the inset. The figure was adapted from Liao et al. (2011).**

4. Referee comment: "*Please state and demonstrate time response, and preferably show a*

*calibration and a zero. Detection limits are given for 2.5 min. I don't understand the relation between the 2.5 min back- ground and the 0.04 s duty cycle for each mass (line 946). Please describe the measurement frequency - how often is a measurement made? I'm confused as to whether these are 0.04 s measurements, 2.5 min measurements, or 1 hr measurements (as used in the figures). And please be specific about the integration time for the mixing ratios. For example, are the range of mixing ratios in the abstract (line 22) for 1 hr averages?"*

**Author response:** The dwell time for each m/z ion was set to 0.5 s and measurements of these ions were obtained every ~13 s. These measurements were averaged to generate 1-hour averaged mixing ratios of the various gases. Mixing ratios presented in this paper are the 1-hour averaged mixing ratios. We have added details on the frequency of measurements and stated explicitly that the data reported are 1-hour averages in the revised manuscript:

**Page 7 line 192: "Ions monitored during the field study included m/z 45, 59, 65, 73, 75, 79, 82, 87, 89, 101, 102, 103, 108, 117, 131 and 148. The assignment of these ions will be discussed in section 3. The dwell time for each m/z ion was set to 0.5 s and measurements of these ions were obtained every ~13 s, which resulted in a ~4 % (= 0.5/13 x 100 %) duty cycle for each ion monitored. The data presented in this paper was averaged to 1-hour intervals unless stated otherwise."**

The author is correct in stating that the range of mixing ratios reported in the abstract are the 1-hour averaged values. This is clarified in the revised manuscript:

**Page 1 line 21: "1-hour averaged ambient concentrations of organic acids ranged from a few parts per trillion by volume (ppt) to several parts per billion by volume (ppb)."**

Each background measurement period lasted ~4 min. However, ion signals for the different organic acids took up to 1.5 min to stabilize during the switch between ambient, calibration and background measurements during the field study. Hence, ion signals obtained during the first 1.5 min were not included in the calculation of the average and standard deviation of ion signals measured during background mode. This resulted in a 2.5 min (= 4 min – 1.5 min) integration time period for background measurements. The duty cycle (0.04 or 4 %) was calculated from the dwell time of each m/z ion monitored (0.5 s) and the measurement frequency (13 s). We clarified how detection limits were calculated from the background measurements in the revised manuscript:

**Page 9 line 257: "The detection limits of the organic acids were estimated as 3 times the standard deviation values (3σ) of the ion signals measured during background mode. Although each background measurement period lasted ~4 min, ion signals of the different organic acids took up to 1.5 min to stabilize during the switch between ambient, calibration and background measurements during the field study. Thus, ion signals measured during the first 1.5 min were not included in the calculation of the average and standard deviation of ion signals measured during background mode. Table 1 summarizes the average detection limits of the organic acids for 2.5 min averaging periods which corresponds to the length of a background measurement with a 4 % duty cycle for each m/z."**

As requested, we have added a section discussing the CIMS instrument time response and a figure showing a calibration and background measurement in the revised manuscript:

**Page 14 line 426: "3.1.3. Background and calibration measurements**

**Figure S4 shows an example of the CIMS instrument response during the switch between background, calibration and ambient measurements of formic and acetic acids during the field study. The 13 s time resolution data was used to determine the CIMS instrument time response. Formic (m/z 45, 65 and 108) and acetic (m/z 79) acid ion signals took ~1.5 min to reach a steady state after switches between ambient, calibration and background measurements (Figs. S4a and S4c).**

**The decays in the formic and acetic acid ion signals and times required for them to reach steady state after the removal of calibration gases during the switch from standard addition calibration to ambient sampling were used to determine the CIMS response time. The signal decays were fitted using double exponential functions. For formic acid, the m/z 45, 65 and 108 ion signals decayed to $1/e^2$ in $37 \pm 2$, $33 \pm 2$ and $32 \pm 2$ s, respectively (Fig. S4b). For acetic acid, the m/z 79 ion signal decayed to $1/e^2$ in $42 \pm 2$ s (Fig. S4d)."**

[Figure]

**Figure S4: Example of the CIMS instrument response during switches between background, calibration and ambient measurements of (a) formic, and (c) acetic acids. Panels (b) and (d) show the percent of formic and acetic acid ion signals after the removal of a 6.75 ppb of formic acid and 5.87 ppb of acetic acid standard addition calibration as a function of time. All the data shown here are 13 s time resolution data. Double exponential fits to each m/z ion are shown as colored solid lines. Black dashed lines show the times for the ions to decay to $1/e^2$.**

5. Referee comment: "*Detection limits and abundances in the figures are written in ppb, but for most cases, ppt would be a more appropriate unit and much easier to read.*"

**Author response:** We have made the requested changes in the revised manuscript.

6. Referee comment: "*Giving sensitivities relative to 34SO2 is confusing and seems unnecessary, especially since that is a different ion-molecule reaction than all the organic acid reactions studied here and it has a water dependence. Providing absolute sensitivities will be much more useful for anyone who wants to compare this ion chemistry with others. And please discuss how these sensitivities compare to other techniques.*"

**Author response:** As requested, we have provided the absolute sensitivities of organic acids and discussed how these sensitivities compare with other CIMS techniques in the revised manuscript:

**Page 12 line 363: "Table 1 shows a summary of the sensitivities of $X^-$ and $X^-\cdot HF$ ions of organic acids. The average sensitivities of the $HCOO^-$ (m/z 45) and $HCOO^-\cdot HF$ (m/z 65) ions of formic acid were 1.29 ± 0.22 and 0.29 ± 0.05 Hz ppt$^{-1}$, respectively, while the average sensitivities of the $CH_3COO^-$ (m/z 59) and $CH_3COO^-\cdot HF$ (m/z 79) ions of acetic acid were 1.46 ± 0.29 and 0.30 ± 0.06 Hz ppt$^{-1}$, respectively. A weak $^{210}Po$ ion source (< 1 mCi) was used by $SF_6^-$-CIMS instrument during the field study, hence these sensitivities will be substantially higher if a stronger radioactive source is used. Nevertheless, these sensitivities are compared to formic and acetic acid sensitivities measured by a high-resolution time-of-flight chemical ionization mass spectrometer (Aerodyne Research Inc.) that utilized $I^-$ reagent ions during the field study. Although the formic acid sensitivity measured by $I^-$-CIMS (1.33 ± 0.28 Hz ppt$^{-1}$) was comparable to that measured by $SF_6^-$-CIMS, the acetic acid sensitivity measured by $I^-$-CIMS (< 0.1 Hz ppt$^{-1}$) was substantially lower than that measured by $SF_6^-$-CIMS. Previous studies have similarly reported low acetic acid sensitivity measured by $I^-$-CIMS (Aljawhary et al., 2013; Lee et al., 2014)."**

**References:**

**Aljawhary, D., Lee, A. K. Y., and Abbatt, J. P. D.: High-resolution chemical ionization mass spectrometry (ToF-CIMS): application to study SOA composition and processing, Atmospheric Measurement Techniques, 6, 3211-3224, 10.5194/amt-6-3211-2013, 2013.**

**Lee, B. H., Lopez-Hilfiker, F. D., Mohr, C., Kurten, T., Worsnop, D. R., and Thornton, J. A.: An Iodide-Adduct High-Resolution Time-of-Flight Chemical-Ionization Mass Spectrometer: Application to Atmospheric Inorganic and Organic Compounds, Environmental Science & Technology, 48, 6309-6317, 10.1021/es500362a, 2014.**

**Table 1: Summary of organic acids of interest, their detection limits and sensitivities of their $X^-$ and $X^-\cdot HF$ ions[a]**

| Organic Acid | Detection limit (ppt)[b] | Sensitivity (Hz ppt$^{-1}$) | |
| --- | --- | --- | --- |
| | | $X^-$ | $X^-\cdot HF$ |
| **Formic acid** | 30 | 1.29 ± 0.22 | 0.29 ± 0.05 |
| **Acetic acid** | 60 | 1.46 ± 0.29 | 0.30 ± 0.06 |
| **Oxalic acid** | 1 | 6.38 ± 0.32 | 0.97 ± 0.05 |
| **Butyric acid** | 30 | 0.41 ± 0.01 | 0.12 ± 0.004 |
| **Glycolic acid** | 2 | 5.53 ± 0.11 | 1.64 ± 0.03 |
| **Propionic acid** | 6 | 2.05 ± 0.02 | 1.26 ± 0.01 |
| **Valeric acid** | 10 | 0.76 ± 0.008 | 0.35 ± 0.004 |

**[a]Only organic acids with calibration measurements are shown.**
**[b]Detection limits are approximated from 3 times the standard deviation values ($3\sigma$) of the ion signals measured during background mode. Shown here are the average detection limits of the organic acids for 2.5 min averaging periods which corresponds to the length of a background measurement at a 4 % duty cycle for each mass.**

7. Referee comment: "*Please discuss the many possible interferences to the organic acids listed.*

*The discussion of calibrations for many compounds was valuable, but then compounds for which calibrations weren't successful (pyruvic, for example) were dismissed. It appears arbitrary to ignore compounds if the calibration didn't work. Those other compounds almost certainly contribute to the signals and make the calculations of WSOC fraction extremely unreliable. For example, the signals at oxalic likely are dominated by or include contributions from lactic acid (both mass 90), butyric includes contributions from pyruvic (both mass 88), propionic includes contributions from glyoxylic acid (both mass 74). The authors correctly note that the mass assignments are speculative. It is OK to speculate, but from then on it would be appropriate to report mixing ratios only as upper limits for that compound and then forego calculations that rely on quantifying abundance. The sensitivities of the reported compounds span a wide range (over an order of magnitude). Thus, if the signal at one of the reported masses comes instead from an isomer with a much higher or lower sensitivity, the organic acid abundance would be drastically over or under estimated. There needs to be considerably more work to justify the WSOC contributions.*"

**Author response:** The comparison of the total carbon contributed by the organic acids to the $WSOC_g$ serves as a zeroth order check that the assignment of ion peaks to the different organic acids are plausible. We agree that the assignment of some ion peaks are speculative, hence we are more circumspect in our discussion of the carbon mass fraction of $WSOC_g$ comprised of organic acids in the revised manuscript:

**Page 19 line 573: "To estimate the fraction of $WSOC_g$ that is comprised of organic acids, the total organic carbon contributed by formic, acetic, oxalic, butyric, glycolic, propionic and valeric acids is compared to the $WSOC_g$ measurements. We emphasize that the ion peak assignment of some of these organic acids are speculative. Hence, this comparison primarily serves as a zeroth order check to determine if the peak assignments are plausible. Figures 6a and 6b show the time series and diurnal profiles of $WSOC_g$ and the organic carbon contributed by the measured organic acids. Formic and acetic acids comprised majority of the total organic carbon contributed by the measured organic acids (study averages of 31 and 38 %, respectively). Assuming that the ion peak assignments are correct and the measured organic acids are completely water-soluble, the carbon mass fraction of $WSOC_g$ comprised of these organic acids ranged from 7 to 100 %."**

**Figure 6: (a) Time series of $WSOC_g$ and the total organic carbon contributed by the measured organic acids. All the data are displayed as 1-hour averages. (b) Diurnal profiles of $WSOC_g$ and the total organic carbon contributed by the measured organic acids. Also shown are the diurnal profiles of the organic carbon contributed by the individual measured organic acids. All the concentrations represent the mean hourly averages and the standard errors are plotted as error bars. (c) Scatter plot of total organic carbon contributed by the measured organic acids with $WSOC_g$. Note that the ion peak assignment to some of these organic acids are speculative.**

8. Referee comment: "*I don't understand the count rates and normalization shown in figures 6-8. How do the count rates get to be small fractions of 1 Hz? Please use a unit that is more accurate and describe how the values are determined.*"

**Author response:** The data for malonic, succinic and glutaric acids are presented as the ratio of their ion signals (Hz) to the instrument's sensitivity to $F_2{}^{34}SO_2$ (Hz ppb$^{-1}$) since these organic acids were not calibrated. The ion signals of these organic acids are 1 to 2 orders of magnitude smaller than the sensitivity of the $F_2^{34}SO_2^-$ ion (study-averaged sensitivity = 2928 ± 669 Hz ppb$^{-1}$), resulting in these ratios to be less than 1. We have changed the units in the figures and stated more explicitly how these values were obtained in the revised manuscript:

**Page 18 line 551: "The time series of ion signals (Hz) of malonic, succinic and glutaric acids normalized to the instrument's sensitivity to $F_2^{34}SO_2$ (Hz ppb$^{-1}$) are shown in Fig. S3. The ion signals of these organic acids are 1 to 2 orders of magnitude smaller than the sensitivity of the $F_2^{34}SO_2^-$ ion (study-averaged sensitivity = 2928 ± 669 Hz ppb$^{-1}$), resulting in these ratios to be less than 1."**

[revised manuscript text omitted]

9. Referee comment: "*I don't understand the reagent ion signal and the use of sulfur isotopes. 34SF6- is said to be the reagent ion (line 338). Is isotopically labeled SF6 used? Why do the magnitudes of the peaks at mass 145 and 147 appear to be similar in Figure 2? Please clearly state the magnitude of the reagent ion signal. From figure 2, it appears that the reagent ion signal is approximately 400 kHz, and similar in magnitude to SF5-, CO3-. Wouldn't ion chemistry from those other ions also contribute to the reactions? A more thorough description of the reagent ion is necessary to understand how the ion chemistry can be dominated by SF6-. Figure 2 makes it appear that other ions could be substantial contributors to the ion chemistry.*"

**Author response:** Isotopically labeled SF₆⁻ was not used as the reagent ion. ³²SF₆⁻ (at m/z 146) is the reagent ion. The signal of the reagent ion $^{32}SF_6^-$ was saturated for the entire field study, which led to the sharp drop in the ion signal at m/z 146 as shown in Fig. 2. Hence, we monitored the ion signal of its isotope $^{34}SF_6^-$ at m/z 148 (which was not saturated) to determine if the reaction of $SF_6^-$ with ambient water vapor and $O_3$ depleted $SF_6^-$ reagent ions. Despite fluctuations in ambient water vapor and $O_3$ concentrations, the $^{34}SF_6^-$ ion signal was relatively constant for the entire field study with a standard deviation of < 3%. This indicates that the reaction of $SF_6^-$ with ambient water vapor and $O_3$ do not deplete $^{32}SF_6^-$ reagent ions.

The ion signal of $^{34}SF_6^-$ (at m/z 148) and the isotopic abundances of $^{32}S$ vs. $^{34}S$ can be used to estimate the theoretical ion signal of the $^{32}SF_6^-$ reagent ion at m/z 146 (i.e., ion signal obtained if no signal saturation occurred). We find that the $^{32}SF_6^-$ reagent ion would have a theoretical ion signal of ~9 x $10^6$ Hz, which is 2 orders of magnitude larger than the $CO_3^-$ signal. In addition, we expect the $SF_6^-$ ion chemistry to dominate since $SF_5^-$ is much more stable than $SF_6^-$ and in any case will likely give similar product ions.

We have revised the manuscript and modified Fig. 2's caption to eliminate any confusion regarding the reagent ion used and contributions of other ions such as $SF_5^-$ and $CO_3^-$ to the ion chemistry:

**Page 13 line 395: "Figure 2 shows a mass spectrum of ambient air. Interference peaks at m/z 39 (F$^-$•(HF) and CO$_3^-$, respectively) can be attributed to the presence of water and O$_3$, respectively. The reagent ion $^{32}$SF$_6^-$ is present at m/z 146. The $^{32}$SF$_6^-$ reagent ion signal was saturated, and this caused the sharp drop in the m/z 146 signal as shown in Fig. 2. Since the $^{32}$SF$_6^-$ reagent ion signal was saturated for the entire field study, we monitored the ion signal of its isotope $^{34}$SF$_6^-$ to determine if the reaction of SF$_6^-$ with ambient water vapor (5.92 x 10$^{-6}$ to 2.19 x 10$^{-5}$ g cm$^{-3}$) and O$_3$ (2.1 to 82.4 ppb) depleted SF$_6^-$ reagent ions. Figure S2a shows the time series of the $^{34}$SF$_6^-$ ion signal and ambient water vapor concentration for the entire field study. Despite fluctuations in ambient water vapor and O$_3$ concentrations, the $^{34}$SF$_6^-$ ion signal was relatively constant for the entire field study with a standard deviation of < 3%. This indicates that the reaction of SF$_6^-$ with ambient water vapor and O$_3$ did not significantly deplete the $^{32}$SF$_6^-$ reagent ions during the field study."**

**Figure 2: Mass spectrum of ambient air and background measured in Yorkville, Georgia on 8 Sept 2016 using SF$_6^-$. Note that the $^{32}$SF$_6^-$ reagent ion signal (at m/z 146) is saturated, causing the sharp drop in its signal. As a result, the ion signal of its isotope $^{34}$SF$_6^-$ (at m/z 150) was monitored to determine if the reaction of SF$_6^-$ with ambient water vapor and O$_3$ depleted SF$_6^-$ reagent ions.**

10. Referee comment: "*Since the major measurement technique advance here is using SF6 in high water and ozone environments, there should be greater discussion of water and ozone mixing ratios in the main text, and the necessary modifications to make the ion chemistry work in this environment. Although the supplement shows some ozone and water data, simpler discussion of the range of ozone and water mixing ratios and the effect on the ion chemistry should be included in the manuscript. Please show how formic or acetic sensitivity varies with ozone and water.*"

**Author response:** Ambient water vapor concentrations ranged from 5.92 x $10^{-6}$ to 2.19 x $10^{-5}$ g cm$^{-3}$, while ambient $O_3$ concentrations ranged from 2.1 to 82.4 ppb during the study. The sampled air was diluted in the CIMS flow tube and the flow tube was operated at a low pressure to minimize interferences by water vapor and $O_3$. Our results indicate that the reaction of $SF_6^-$ with ambient water vapor and $O_3$ did not deplete the $^{32}SF_6^-$ reagent ions. The $SO_2$ sensitivity showed a clear linear relationship to water vapor and varied within a factor of two for the entire field study, while the $SO_2$ sensitivity did not show any obvious dependence on ambient $O_3$ concentrations. The formic and acetic acid sensitivities did not show any obvious dependence on ambient water vapor and $O_3$ concentrations. Therefore, we do not expect the sensitivities of the other organic acids to depend on ambient water vapor and $O_3$ concentrations.

As requested, we have added more discussion about the ambient water vapor and $O_3$ concentrations and their effects on $SF_6^-$ ion chemistry, the necessary sampling configurations needed to make $SF_6^-$ ion chemistry to work in the field study, and how the formic and acetic acid sensitivity varies with water vapor and $O_3$:

**Page 13 line 398: "Since the $^{32}SF_6^-$ reagent ion signal was saturated for the entire field study, we monitored the ion signal of its isotope $^{34}SF_6^-$ to determine if the reaction of $SF_6^-$ with ambient water vapor (5.92 x 10$^{-6}$ to 2.19 x 10$^{-5}$ g cm$^{-3}$) and $O_3$ (2.1 to 82.4 ppb) depleted $SF_6^-$ reagent ions. Figure S2a shows the time series of the $^{34}SF_6^-$ ion signal and ambient water vapor concentration for the entire field study. Despite fluctuations in ambient water vapor and $O_3$ concentrations, the $^{34}SF_6^-$ ion signal was relatively constant for the entire field study with a standard deviation of < 3%. This indicates that the reaction of $SF_6^-$ with ambient water vapor and $O_3$ did not significantly deplete the $^{32}SF_6^-$ reagent ions during the field study.**

**The $F_2^{34}SO_2^-$ ion signal was used to monitor the CIMS $SO_2$ sensitivity during the field study. Figure S2b shows the time series of the $F_2^{34}SO_2^-/^{34}SF_6^-$ ion signal ratio obtained in calibration measurements. There is a noticeable increase in the $F_2^{34}SO_2^-/^{34}SF_6^-$ ion signal ratio on 28 Sept 2016, indicating an increase in the CIMS instrument sensitivity. The increase in CIMS instrument sensitivity is due to the decrease in ambient water vapor concentrations on 28 Sept 2016 (Fig. S2a). Previous laboratory and field studies showed that this was due to the hydrolysis of $F_2^{34}SO_2^-$, which led to the loss of this ion and diminished sensitivity at higher levels of ambient water vapor (Arnold and Viggiano, 2001; Slusher et al., 2001). However, the $SO_2$ sensitivity at $F_2^{34}SO_2^-$ only varied within a factor of two for the entire field study with a clear relationship to water vapor (Fig. S2c). The $SO_2$ sensitivity did not show any obvious dependence on ambient $O_3$ concentrations (Fig. S2d).**

**The formic (HCOO$^-$ at m/z 45 and HCOO$^-$•HF at m/z 65) and acetic (CH$_3$COO$^-$•HF at m/z 79) acid ions did not show any obvious dependence on ambient water vapor and $O_3$ concentrations during calibration measurements (Fig. S3). Therefore, we do not expect the sensitivities of the X$^-$ and X$^-$•HF ions of the studied organic acids to depend on ambient water vapor and $O_3$ concentrations. We accounted for water vapor dependence of the $F_2^{34}SO_2^-$ ion signal in our post-field calibrations where the response of the CIMS acid signals were measured relative to the of the $^{34}SO_2$ sensitivity."**

[Figure]

**Figure S2: (a) Time series of $^{34}SF_6^-$ reagent ion signal and ambient water vapor concentration for the entire field study. The ambient water vapor mass concentrations are determined from ambient relative humidities and temperatures. (b) Time series of $F_2{}^{34}SO_2^-/^{34}SF_6^-$ ion signal ratio obtained during calibration measurements. Panels (c) and (d) show the $F_2{}^{34}SO_2^-$ ion sensitivity obtained from calibration measurements as a function of ambient water vapor and $O_3$ concentrations. Data in panels (a) to (d) are displayed as 1-hour averages.**

[Figure]

**Figure S3: Panels (a) and (b) show the sensitivities of formic acid ions (HCOO$^-$ at m/z 45, HCOO$^-$•HF at m/z 65, and SF$_4^-$ at m/z 108) obtained from calibration measurements as a function of ambient water vapor and $O_3$ concentrations. Panels (c) and (d) show the acetic acid sensitivity (CH$_3$COO$^-$•HF at m/z 79) obtained from calibration measurements as a function of ambient water vapor and $O_3$ concentrations.**

11. Referee comment: "*One overlooked feature in the results is the differing diurnal variations of*

*acetic and formic acids in figures 3-4. Rather than focus on organic acid budgets that rely on speculative assignments and suffer from unexamined interferences, a study of the diurnal variability of acetic and formic acids could prove interesting.*"

**Author response:** We do not know what caused the observed differing diurnal variations of formic and acetic acids. We are currently employing laboratory and modeling studies to explain these field observations. Results of these studies will be the subject of future papers.

**Response to Referee 2 (Referees' comments are italicized)**

1. Referee comment: "*The authors make a strong statement (line 95) that new techniques for the real-time measurement of gas-phase organic acids are need due to deficiencies in existing CIMS based ion chemistry (acetate and iodide CIMS). The authors cite issues with acetate CIMS in the detection of acetic acid and the wide range of sensitivities to different organic acids in iodide CIMS. After reading this paper it was not clear to me that SF6- has an advantage over these techniques. It was shown that interferences due to O3 hinder detection of acetic acid and there is an order of magnitude spread in sensitivity to various organic acids in Table 1. I think the authors need to better articulate how this technique is an advance over existing ion chemistry or acknowledge that it is a parallel approach to existing ion chemistry.*"

**Author response:** The $SF_6^-$-CIMS technique serves as alternative approach to existing CIMS techniques in the detection of organic acids. The major advantage that $SF_6^-$ has over $I^-$ and $CH_3CO_2^-$ is that it allows for the detection of acetic acid and $SO_2$. $CF_3O^-$ has a similar chemistry to $SF_6^-$ but it also has issues due to hydrolysis and the ion precursor is not commercially available. This information has been added in the revised manuscript:

**Page 4 line 107: "The sulfur hexafluoride ($SF_6^-$) anion has been used as a CIMS reagent ion to measure atmospheric inorganic species such as sulfur dioxide ($SO_2$), nitric acid ($HNO_3$) and peroxynitric acid ($HO_2NO_2$) (Slusher et al., 2001; Slusher et al., 2002; Huey et al., 2004; Kim et al., 2007). $SF_6^-$ commonly reacts with most acidic gases at the collision rate by either proton or fluoride transfer reactions (Huey et al., 1995). The $SF_6^-$ ion chemistry is selective to acidic species, which can simplify the mass spectral analysis of organic acids. However, $SF_6^-$ is reactive to both ozone ($O_3$) and water vapor, which can lead to interfering reactions that limit its applicability to many species in certain environments (Huey et al., 2004). For these reasons, this work is focused on assessing the ability of $SF_6^-$ to measure a series of organic acids in ambient air. The major advantage that $SF_6^-$ has over $I^-$ and $CH_3CO_2^-$ is that it allows for the detection of acetic acid and $SO_2$. $CF_3O^-$ has a similar chemistry to $SF_6^-$ but it also has issues due to hydrolysis and the ion precursor is not commercially available."**

**Author response:** We chose a 4 min background measurement time to give the scrubber (during background measurements) and flow tube ample time to equilibrate when the three-way PFA Teflon valve was switched between ambient and background modes, and to obtain good averaging statistics. This is stated explicitly in the revised manuscript:

**Page 7 line 204: “Each background and calibration measurement period lasted ~4 and ~3.5 min, respectively, which not only gave the scrubber (during background measurements) and flow tube ample time to equilibrate when the three-way PFA Teflon valve was switched between ambient and background modes, but also allowed us to obtain good averaging statistics during background and calibration measurements.”**

We also added a section (3.1.3) and a figure (Fig. S4) discussing instrument time response during background and calibration measurements in the revised manuscript. Please refer to our response to comment 4 from referee 1.

7. Referee comment: “*Line 183: What was the 1.12 ppm SO2 standard diluted to? Presumably calibrations were not done at this mixing ratio.*”

**Author response:** 1.85 ppb of ³⁴SO₂ was used in calibration measurements. This information has been added to the revised manuscript:

**Page 7 line 209: “1.85 ppb of ³⁴SO₂ was added to sampled air flow during calibration measurements.”**

8. Referee comment: “*Line 184: Again, it would be more helpful to present as the concentration of FA or AA that is delivered instead of the permeation tube emission rates.*”

**Author response:** As requested, this information has been added to the revised manuscript:

**Page 7 line 216: “6.75 ppb of formic acid and 5.87 ppb of acetic acid was added to sampled air flow during calibration measurements.”**

9. Referee comment: “*Line 308: For reactions 1a-c, should one think of these as separate reaction channels governed by ion-molecule kinetics or does every reaction proceed through 1a and the electric field strength of the CDC sets the ratio of the observed products. This may lead to strong*

*deviations in the observed products based on instrument configuration.*"

**Author response:** Every reaction of $SF_6^-$ with organic acids (HX) proceeds through reactions 1a to 1c, where the ratio of the observed product ions can be controlled by the field strength of the CDC. This is stated explicitly in the revised manuscript:

**Page 11 line 331: "CIMS measurements of atmospheric constituents use ion-molecule reactions to selectively ionize compounds of interest in the complex matrix of ambient air and produce characteristic ions. The reactions of $SF_6^-$ with the organic acids (HX) proceed through reactions 1a to 1c, and gave similar products to those reported previously for $SF_6^-$ reactions with inorganic acids (Huey et al., 1995): $SF_5^-$, $X^-$ and $X^-•HF$ where $X^-$ is the conjugate base of the organic acid (reactions 1a-c).**

$$SF_6^- + HX \rightarrow X^-•HF + SF_5 \qquad\qquad\qquad \textbf{(1a)}$$

$$SF_6^- + HX \rightarrow X^- + HF + SF_5 \qquad\qquad\qquad \textbf{(1b)}$$

$$SF_6^- + HX \rightarrow SF_5^- + HF + X \qquad\qquad\qquad \textbf{(1c)}$$

**The effective branching ratios of the $SF_5^-$, $X^-$ and $X^-•HF$ product ions can be impacted by the field strength of the CDC."**

**Author response:** The ratio of the sampled air to the $N_2/SF_6$ mixture introduced into the CIMS flow tube is approximately 1:13. The referee is correct in pointing out that the reduction in ion-molecule reaction time helps minimizes interfering secondary ion chemistry. We reduced the ion-molecule reaction time by sampling only 0.3 L min$^{-1}$ of air through the variable orifice into the flow tube and maintaining the flow tube at a low pressure (~13 mbar). This information is stated explicitly in the revised manuscript.

**Page 13 line 384: "These reactions can deplete $SF_6^-$ as well as form a variety of potentially interfering ions from secondary reactions (e.g., $F^-•(HF)_n$ and $CO_3^-$ ions) that depend on more abundant atmospheric species. For these reasons, efforts were made to minimize interferences by limiting reaction times and the flow sampled into the CIMS. This was accomplished by sampling only 0.3 L min$^{-1}$ of air through the variable orifice into the flow tube and maintaining the flow tube at a low pressure (~13 mbar). The 0.3 L min$^{-1}$ sampled air flow is diluted by 3.7 slpm of $N_2/SF_6$ flow in the flow tube. The ratio of the sampled air flow to the $N_2/SF_6$ flow introduced into the flow tube is approximately 1:13. While the high**

**N$_2$/SF$_6$ flow (3.7 slpm) passed through the radioactive source into the flow tube increased the SF$_6^-$ reagent ion signal, the high dilution of the sampled air flow in the flow tube reduced the CIMS instrument sensitivity by decreasing the number density of the analytes.”**

11. Referee comment: “*Line 335: What is the uncertainty in the IMR and CDC pressure? Are these pressures also controlled?*”

**Author response:** The uncertainty in the flow tube pressure (the so-called IMR) is 0.5 %, while the uncertainty in the CDC pressure is 10 %. These pressures are controlled by a variable orifice. This information has been added to the revised manuscript:

**Page 6 line 176: “Since these F$^-$•(HF)$_n$ cluster ions could interfere with mass spectral analysis, the flow tube was maintained at a low pressure (~13 mbar, 0.5 % uncertainty) in this study to minimize interferences from SF$_6^-$ reaction with water vapor. The analyte ions exited the flow tube and were accelerated through the collisional dissociation chamber (CDC), which was maintained at ~0.8 mbar (10 % uncertainty). The molecular collisions in the CDC served to dissociate weakly bound cluster ions into their core ions to simplify mass spectral analysis. Flow tube and CDC pressures were controlled by the automatic variable orifice.”**

**The following are additional minor changes the authors have made to the manuscript:**

1. We corrected the model series of the CIMS turbo pumps:

**Page 6 line 168: “These sections were evacuated by a scroll pump (Edward nXDS 20i), a drag pump (Adixen MDP 5011) and two turbo pumps (Varian Turbo-V301), respectively.”**

2. We corrected the uncertainty in nitric acid ambient concentrations:

[revised manuscript text omitted]
 and $O_3$ concentrations. Data in panels (a) to (d) are displayed as 1-hour averages.

[Figure]

[Figure]

**Figure S3:** Panels (a) and (b) show the sensitivities of formic acid ions (HCOO$^-$ at m/z 45,

HCOO$^-$•HF at m/z 65, and SF$_4^-$ at m/z 108) obtained from calibration measurements as a function of ambient water vapor and O$_3$ concentrations. Panels (c) and (d) show the acetic acid sensitivity (CH$_3$COO$^-$•HF at m/z 79) obtained from calibration measurements as a function of ambient water vapor and O$_3$ concentrations.

[Figure]

**Figure S4:** Example of the CIMS instrument response during switches between background, calibration and ambient measurements of (a) formic, and (c) acetic acids.

Panels (b) and (d) show the percent of formic and acetic acid ion signals after the removal of a 6.75 ppb of formic acid and 5.87 ppb of acetic acid standard addition calibration as a function of time. The data shown here is 13 s time resolution data. Double exponential fits to each m/z ion are shown as colored solid lines. Black dashed lines show the times for the ions to decay to $1/e^2$.

[Figure]

[Figure]

**Figure S5:** Time series of signals of (a) malonic, (c) succinic, and (e) glutaric acids measured during the field study. The data are displayed as 1-hour averages. Their corresponding diurnal profiles are shown in (b), (d) and (f), respectively. All the signals represent averages in 1-hour intervals and the standard errors are plotted as error bars.

These organic acids were not calibrated so all the signals are presented here as Hz normalized by the instrument's sensitivity to $F_2{}^{34}SO_2$ (Hz ppb$^{-1}$), which was the primary calibrant used in the field study.

[Figure]

**Figure S6:** Time series of (a) SO$_2$ and (b) HNO$_3$ concentrations measured during the field study. All the data are displayed as 1-hour averages.

[Figure]

[Figure]

**Figure S7:** Scatter plots of concentrations (or signals) of (a) formic, (b) acetic, (c) oxalic, (d) butyric, (e) glycolic, (f) propionic, (g) valeric, (h) malonic, (i) succinic, and (j) glutaric acids with CO concentration. All the data are displayed as 1-hour averages. The data for malonic, succinic and glutaric acids are presented as the ratio of their ion signals (Hz) to the instrument's sensitivity to $F_2{}^{34}SO_2$ (Hz ppb$^{-1}$) since these organic acids were not calibrated. Red lines shown are linear fits to the data.

[Figure]

[Figure]

**Figure S8:** Scatter plots of concentrations (or signals) of (a) formic, (b) acetic, (c) oxalic, (d) butyric, (e) glycolic, (f) propionic, (g) valeric, (h) malonic, (i) succinic, and (j) glutaric acids with $SO_2$ concentration. All the data are displayed as 1-hour averages. The data for malonic, succinic and glutaric acids are presented as the ratio of their ion signals (Hz) to the instrument's sensitivity to $F_2{}^{34}SO_2$ (Hz ppb$^{-1}$) since these organic acids were not calibrated. Red lines shown are linear fits to the data.

[Figure]

[Figure]

**Figure S9:** Scatter plots of concentrations (or signals) of (a) formic, (b) acetic, (c) oxalic, (d) butyric, (e) glycolic, (f) propionic, (g) valeric, (h) malonic, (i) succinic, and (j) glutaric acids with $O_3$ concentration. All the data are displayed as 1-hour averages. The data for malonic, succinic and glutaric acids are presented as the ratio of their ion signals (Hz) to the instrument's sensitivity to $F_2^{34}SO_2$ (Hz ppb$^{-1}$) since these organic acids were not calibrated. Red lines shown are linear fits to the data.

[Figure]

[Figure]

**Figure S10:** Scatter plots of concentrations (or signals) of (a) formic, (b) acetic, (c) oxalic, (d) butyric, (e) glycolic, (f) propionic, (g) valeric, (h) malonic, (i) succinic, and (j) glutaric acids with $HNO_3$ concentration. To exclude periods when the site was affected by urban or power plant emissions, data where $HNO_3 > 0.5$ ppb are excluded from these scatter plots. All the data are displayed as 1-hour averages. The data for malonic, succinic and glutaric acids are presented as the ratio of their ion signals (Hz) to the instrument's sensitivity to $F_2^{34}SO_2$ (Hz ppb$^{-1}$) since these organic acids were not calibrated. Red lines shown are linear fits to the data.

[Figure]

**Figure S11:** (a) Time series of isoprene concentration during the field study. (b) Diurnal profile of isoprene. All the concentrations represent averages in 1-hour intervals and the standard errors are plotted as error bars. (c) Scatter plot of isoprene concentration with ambient temperature. All the data are displayed as 1-hour averages.

[Figure]

**Figure S12:** Scatter plots of concentrations of (a) formic and (b) acetic acids with isoprene concentration. All the data are displayed as 1-hour averages. Red lines shown are linear fits to the data.

---

## Author Comment (AC2) · 24 May 2018

We greatly value the careful reading and the detailed comments provided by the referees. The responses to the comments of the referees in our direct reply and within the revised manuscript are provided in the attached pdf. The pages and lines indicated correspond to those in the marked copy.

Please also note the supplement to this comment:
https://www.atmos-meas-tech-discuss.net/amt-2018-46/amt-2018-46-AC2-supplement.pdf

---

## Author Response (AR2)

The responses to the comments of the referee in our direct reply (shown below) and within the revised manuscript (see marked copy) are provided. The pages and lines indicated below correspond to those in the marked copy.

**Response to Referee 1 (Referee's comments are italicized)**

1. Referee's comment: "*Although several caveats have been introduced noting that the peak assignments are speculative, the same interpretations follow. The speculation is ok, but there should be no further quantification or interpretation based upon that. Without further justification, the organic acids should not be identified as a single compound and their abundance should not be interpreted. Instead, please show them as masses, or as a sum of compounds. The figures may be very misleading by showing individual compounds as mixing ratios. If molecule names are used, then some simple tests ought to be performed to inform the identification of the masses. Can an HR-TOF be used with SF6- in a similar environment to see which compounds dominate? That would be far more powerful than the comparison with WSOCg.*"

**Author response:** We will agree that that with the exception of formic, acetic, oxalic and propionic acids, we do not have sufficient evidence to justify the assignment of other the organic acids. Hence, we will refer to organic acids with m/z 75, 87, 101, 103, 117 and 131 by their ion masses.

We are confident in our peak assignment of formic, acetic and propionic acids because the $SF_6^-$ ion chemistry is selective to acidic species and these three acids do not have organic acid isomers and isobaric species. In addition, we are confident in our peak assignment of oxalic acid because the estimated gas-phase oxalic acid concentration make sense relative to its particle concentrations, which were also measured during the study. As explained in Nah et al. (2018), the gas-particle ratios of the organic acids depend of their thermodynamic conditions, which are dependent on the acid's physicochemical properties, ambient temperature, particle water and pH. In the case of oxalic acid, the measured gas-particle partitioning ratios are in good agreement with their corresponding thermodynamic predictions, thus indicating that our assignment of the m/z 89 ion to oxalic acid is reasonable.

The manuscript has been revised as follows:

**Page 18 line 546: "3.2.3. Larger organic acids**

**In addition to formic and acetic acid, eight other ions were monitored during the field study: m/z 73, 75, 87, 89, 101, 103, 117 and 131. These ions were chosen as they had significant signals when ambient air was sampled and were not obviously formed from $SF_6^-$ reaction with water vapor or $O_3$. Since the CIMS utilized in this study only had unit mass resolution, these ions are the sum of all organic acid isomers and isobaric organic acids of the same molecular weight as well as other product ions from species that might react with $SF_6^-$. We will refer to organic acids with m/z 75, 87, 101, 103, 117 and 131 by their ion masses. We assign the m/z 73 ion as the $X^-$ ion of propionic acid because it does not have organic acid isomers and isobaric species at that m/z. In addition, real-time ion chromatography measurements of aerosol composition performed during the field study demonstrated the presence of particulate oxalic acid (Nah et al., 2018). For this reason, we assign the m/z 89 ion as the $X^-$ ion of oxalic acid. As shown in Nah et al. (2018), the gas-particle ratios of the**

[revised manuscript text omitted]

2. Referee's comment: "*Comparison to aerosol composition (section 3.2.3) is not a compelling reason to choose the peak assignments. As is stated in the manuscript, the low volatility compounds measured in the particle phase are expected to have low gas phase concentrations. This could be used to argue that oxalic, malonic... that are present in the particle phase do not explain the signals measured in the CIMS. The IC aerosol measurements are never discussed quantitatively: Do the gas phase concentrations estimated here make sense relative to the particle concentrations?*"

**Author response:** As the IC aerosol measurements have been discussed in detail in another paper, they were not discussed quantitatively in this paper. We refer the referee to Nah et al. (2018) for detailed discussions of the IC aerosol measurements.

Although volatility is a factor, the gas-particle ratios of the organic acids is more complex than that. It depends on thermodynamic conditions, which are dependent on the acid's physicochemical properties (including volatility), ambient temperature, particle water and pH. As explained in detail in Nah et al. (2018), the measured oxalic acid gas-particle partitioning ratios are in good agreement with their corresponding thermodynamic predictions. Hence, we are confident in our assignment of the m/z 89 ion to oxalic acid since its gas-phase concentrations estimated here make sense relative to its particle concentration. In the case of formic and acetic acids, their measured gas-particle partitioning ratios do not agree with their corresponding thermodynamic predictions, which may be due to mixing states or specific chemical forms of particle-phase formate and acetate (Nah et al., 2018).

Since thermodynamic modeling was not performed for the other organic acids, we are unable to determine if their estimated gas-phase concentrations make sense relative to their particle-phase concentrations. We refer the referee to our reply to comment 1 for revisions made to manuscript regarding the assignment of ions peaks to organic acids.

References:

Nah, T., Guo, H., Sullivan, A. P., Chen, Y., Tanner, D. J., Nenes, A., Russell, A., Ng, N. L., Huey, L. G., and Weber, R. J.: Characterization of Aerosol Composition, Aerosol Acidity and Organic Acid Partitioning at an Agriculture-intensive Rural Southeastern U.S. Site, Atmos. Chem. Phys. Discuss., in review, 10.5194/acp-2018-373, 2018.

3. Referee's comment: "*It would be much more useful to further justify the technique, which may have some advantages over iodide ion chemistry. For example, the sensitivity to propionic acid is much better than iodide. Is the same true for other acids that iodide does not detect well, like acrylic acid? Iodide often suffers from a large lactic acid interference that compromises oxalic acid quantification. What is the sensitivity to lactic acid? Since the motivation was to quantify organic acids, further discussion of the advantages (possibly unique) of SF6 for quantifying organic acids would be particularly valuable.*"

**Author response:** Since the main purpose of this paper is to introduce $SF_6^-$-CIMS as an alternative and promising approach in the detection of organic acids in ambient measurements, the manuscript focuses on the details of the $SF_6^-$ ion chemistry and optimal instrument operation for the detection of such acids. Because this paper is not a comparison paper for the different CIMS techniques on the detection of organic acids, we did not conduct laboratory experiments to compare the $SF_6^-$ vs. $I^-$ sensitivities of different organic acids.

Our preliminary lab experiments showed that $SF_6^-$ is less sensitive to lactic acid compared to oxalic acid because the $SF_6^-$ + lactic acid reaction results primarily in the formation of $SF_5^-$ ions, instead of $X^-$ and $X^-\bullet HF$ ions. As already discussed in the manuscript, the production of $SF_5^-$ does not allow for the selective detection of any atmospheric species. In addition, the larger the branching ratio of the $SF_5^-$ channel, the lower the CIMS sensitivity to an individual acid since the effective rate constants for the $X^-$ and $X^-\bullet HF$ channels are lower. In addition, we always used gloves when working on the CIMS during this study to limit contamination from emissions from human skin. We also kept people away from the front of the $SF_6^-$-CIMS sampling inlet to minimize lactic acid interferences. Furthermore, as noted above, we believe that our assignment of the m/z 89 ion to oxalic acid is reasonable because our estimated gas-phase oxalic acid concentrations make sense relative to its particle concentration. As discussed in detail in Nah et al. (2018), the gas-particle ratios of the organic acids depend of their thermodynamic conditions, which are dependent on the acid's physicochemical properties, ambient temperature, particle water and pH. The measured oxalic acid gas-particle partitioning ratios were shown to be in good agreement with their corresponding thermodynamic predictions. We refer the referee to our reply to comment 1 for revisions made to manuscript regarding the assignment of m/z 89 to oxalic acid.

Given that the main purpose of this paper is to introduce $SF_6^-$-CIMS as an alternative and promising approach in the detection of organic acids in ambient measurements, we feel that we have sufficiently shown the unique advantages of $SF_6^-$-CIMS over other reagent ions. As already stated in the manuscript in the introduction, the major advantage that $SF_6^-$ has over $I^-$ and $CH_3CO_2^-$ is that it allows for the detection of acetic acid and $SO_2$. $CF_3O^-$ has a similar chemistry to $SF_6^-$ and can detect organic acids but it also has issues due to hydrolysis and the ion precursor is not commercially available. Together, our measurements show that $SF_6^-$-CIMS is a promising technique for the simultaneous detection of inorganic species (e.g., $SO_2$, $HNO_3$) and organic acids up to $C_5$ (valeric acid).

References:

Nah, T., Guo, H., Sullivan, A. P., Chen, Y., Tanner, D. J., Nenes, A., Russell, A., Ng, N. L., Huey, L. G., and Weber, R. J.: Characterization of Aerosol Composition, Aerosol Acidity and Organic Acid Partitioning at an Agriculture-intensive Rural Southeastern U.S. Site, Atmos. Chem. Phys. Discuss., in review, 10.5194/acp-2018-373, 2018.

Information regarding how we limited lactic acid contamination during the study has been added to the revised manuscript:

**Page 6 line 184: "It should also be noted that we always used gloves when working on the CIMS during this study to limit contamination of lactic acid emissions from human skin. In addition, we kept people away from the front of the $SF_6^-$-CIMS sampling inlet to minimize lactic acid interferences as well."**

We have added the following sentences to the revised manuscript to emphasize the high sensitivity of $SF_6^-$-CIMS to oxalic, propionic and glycolic acids which are expected to be present at low concentrations in the atmosphere:

**Page 23 line 772: "It should be noted that the $SF_6^-$ CIMS method is particularly sensitive to oxalic, propionic and glycolic acids, which are expected to be present at low concentrations in the atmosphere."**

4. Referee's comment: "*I don't understand the normalization to F2SO2. F2SO2 has a water dependence that the organic acids do not, so the normalization is not appropriate, even if the water dependence is later adjusted (which I didn't understand). Why not normalize to the reagent ion signal, as is common in most other CIMS papers? This would make the work more easily compared to other CIMS techniques, and avoid correcting and then uncorrecting the signals by using F2SO2 normalization.*"

**Author response:** We understand the referee's point regarding the normalization of signals to $F_2SO_2$. While is it true that some research groups do normalize their measured signals to the reagent ion signal, this was not carried out in this paper because the $SF_6^-$ reagent ion signals were in a region of non-linearity (i.e., at or close to signal saturation) for the entire field study. While we could have corrected for the non-linearity in the $SF_6^-$ reagent ion signal prior to normalization, doing so introduces uncertainties to our measurements. As a point of reference, we can only count linearly to a few hundred thousand ions per second and our counter/detector saturates at about a million ions per second. These numbers are not precise and depend on the particular combination of preamplifier and detector. So, we avoid normalization but instead calibrate often or even continuously.

Although we could have normalized our measured CIMS signals to the sensitivities of the formic or acetic acid calibrant gases, the formic and acetic acid sensitivities were not measured for the entire study because their perm tubes were shared with another CIMS during the study. In contrast, $^{34}SO_2$ was the main calibrant gas and its sensitivity was measured during every calibration period for the entire field study. Consequently, the CIMS instrument sensitivity measured by the $F_2^{34}SO_2^-$ ion signal was applied to all the measured species (except for formic and acetic acids during periods when their perm tubes were used for calibration purposes by the $SF_6^-$-CIMS) using relative sensitivities determined in laboratory studies.

5. Referee's comment: "*With the removal of speculation in sections 3.24 and 3.3, there is room to emphasize some of the measurement technique accomplishments. The reagent ion signal is never mentioned in the text, though it is stated in the authors comments. Achieving a signal of 9 MHz is extraordinary given the low activity of the ionizer. Few CIMS do any better than this, so lines 352-354 are not correct in general. The paper should detail how the large reagent ion signal was achieved, as this seems to be a major new advance.*"

**Author response:** We disagree that the $SF_6^-$ reagent ion signal is extraordinarily high and that lines 352 to 354 are incorrect. It may be true that some commercial CIMS can't achieve such high levels of reagent ion signal, but our group has achieved this level of $SF_6^-$ and $I^-$ reagent ion signals ($>10^6$ Hz) with our custom-built CIMS for many years. We refer the referee to Fig. 5 of Slusher et al. (2004) for an example of the $I^-$ reagent ion signal ($\sim 10^6$ Hz) obtained during a previous field study in 2002. In this example, the $I^-$ signal was also saturated at an apparent signal of $10^6$ Hz and was probably more on the order of 10 MHz if it could be counted accurately.

The obtainment of such high levels of reagent ion signals by our CIMS is probably due to the evolution of the design of our CIMS. This has been documented extensively, and we refer the referee to Liao et al. (2011) (or other previous papers from our group) for a more detailed description of our instrument.

**Author response:** We disagree with the referee's assertion on the need for a table that compares the $SF_6^-$ sensitivities of all the organic acids to other reagent ions. The main purpose of this paper is to introduce $SF_6^-$-CIMS as an alternative and promising approach in the detection of organic acids in ambient measurements. This paper is not a comparison paper for the different CIMS techniques on the detection of organic acids. In addition, we think a table comparing sensitivities is likely to be misleading as instruments are constantly developing and being used in different configurations. Hence, the manuscript focuses on the details of the $SF_6^-$ ion chemistry and optimal instrument operation for the detection of such acids.

Given the main purpose of this paper, we strongly feel that we have sufficiently shown the advantages of $SF_6^-$-CIMS over other reagent ions such as $CH_3CO_2^-$, $I^-$ and $CF_3O^-$ to justify it being a promising technique in the detection of ambient organic acids. As already stated in the manuscript and in our previous reply to comment 1 from referee 2, the major advantage that $SF_6^-$ has over $I^-$ and $CH_3CO_2^-$ is that it allows for the detection of acetic acid and $SO_2$. $CF_3O^-$ has a similar chemistry to $SF_6^-$ and can detect organic acids but it also has issues due to hydrolysis and the ion precursor is not commercially available. Together, our measurements show that $SF_6^-$-CIMS is a promising technique for the simultaneous detection of inorganic species (e.g., $SO_2$, $HNO_3$) and organic acids up to $C_5$ (valeric acid).

The referee's comment regarding the $SF_6^-$ sensitivity to lactic acid is addressed in our replies to comment 1 and 3.

7. Referee's comment: "*The 1/e2 time decays are given, but the time response for most of the system is very fast. It looks like 1/e is <13s. Where does the time response come from? It is slower than many other CIMS measurements, even though there is a very large flow through the flow tube, and the inlet appears to be only 25 cm and with a large flow. Do the high dilution or reduced pressure affect time response?*"

**Author response:** Determination of the time response of our CIMS is limited by the CIMS sampling time. As stated in the manuscript, the sampling conditions of the CIMS during the study resulted in 13 s time resolution data. Hence, it is not possible to accurately determine the times for the various signals to decay to 1/e since it appears that the times all appear to be ≤ 13 s. We did not put this in the first version of the paper for these reasons and only did so at the suggestion of a reviewer. In addition, as stated in the manuscript, we report 1-hour averaged ambient concentrations (not 13 s ambient concentrations), which is sufficient for the purpose of this ground-based field study where the concentrations of measured species do not change rapidly (unlike airborne campaigns). Hence, we do not expect the time response of our CIMS to affect our results and conclusions.

However, the high dilution and reduced pressure do not affect the CIMS time response relative to other configurations of our CIMS. Instead, we expect the CIMS time response to a species to be governed primarily by the species' propensity to adhere to surfaces. This information has been added to the revised manuscript:

**Page 15 line 450: "The CIMS time response to a compound is governed primarily by the compound's propensity to adhere to surfaces. The decays in the formic and acetic acid ion signals and times required for them to reach steady state after the removal of calibration gases during the switch from standard addition calibration to ambient sampling were used to determine the CIMS response time."**

8. Referee's comment: "*line 242: what were the impurities in glyoxylic?*"

**Author response:** We were unable to identify the impurities in our glyoxylic acid. Although the purity of the sample was listed as 98 % by the manufacturer (Sigma Aldrich), the impurities were not listed. Some of these m/z peaks appeared at masses larger than glyoxylic acid, suggesting that they may be polymers.

9. Referee's comment: "*line 244: what are the vapor pressures? would heating generate sufficient vapor pressure?*"

**Author response:** The vapor pressures of malonic, succinic and glutaric acids are $5.73 \times 10^{-4}$, $1.13 \times 10^{-4}$ and $4.21 \times 10^{-4}$ kPa at 298 K, respectively (Booth et a., 2010). Although the referee is correct in stating that heating the malonic, succinic and glutaric acid samples will likely generate sufficient vapors for calibration, this method of generating calibrant gases for calibration purposes is subjected to errors caused by the vapors condensing out and adhering onto surfaces at room temperature prior to introduction into the CIMS. It is very difficult to maintain all parts of the tubing, inlet, and CIMS at an elevated temperature which would be required if we used a heated sample. This information has been added to the revised manuscript:

**Page 9 line 249: "We attempted to generate calibration plots for malonic (Sigma Aldrich, ≥ 99.5 %), succinic (Sigma Aldrich, 99 %) and glutaric (Sigma Aldrich, 99 %) acids by passing N$_2$ over their solid samples at room temperature. However, it was not possible to generate large enough gas phase concentrations for calibration since these organic acids have very low vapor pressures. The vapor pressures of malonic, succinic and glutaric acids are $5.73 \times 10^{-4}$, $1.13 \times 10^{-4}$ and $4.21 \times 10^{-4}$ kPa at 298 K, respectively (Booth et al., 2010), which are at least 2 orders of magnitude lower than the organic acids that we calibrated for. Although heating up the malonic, succinic and glutaric acid samples will likely generate sufficient vapors for calibration, this method of generating calibrant gases will lead to large measurement uncertainties due to vapors condensing out and adhering onto surfaces at room temperature prior to introduction into the CIMS."**

**Author response:** The formic acid/CO ratios ranged from $1.0 \times 10^{-3}$ to $2.5 \times 10^{-2}$ ppb ppb$^{-1}$, with an average of $8.7 \times 10^{-3} \pm 5.8 \times 10^{-3}$ ppb ppb$^{-1}$. The CO concentrations were consistently low during the study and ranged from 80 to 240 ppb, with consistent diurnal trends (see Fig. a below). Since the CO concentrations and diurnal profile have been published in another paper (Nah et al. 2018), they were not shown in this manuscript. Based on the diurnal profile of the formic acid/CO ratio (see Fig. b below), the formic acid/CO ratio peaks in the mid-afternoon, which coincides with when formic acid and CO reach their maximum and minimum, respectively.

[Figure]

Given that 1) the CO concentration did not spike during the field study, 2) the formic acid has a consistent diurnal profile, and 3) the formic acid/CO ratio time series is consistent with the diurnal trends of CO and formic acid, it is unlikely that the increase in that the increase in formic acid during the day can be explained by transported CO. This information has been added to the revised manuscript:

**Page 21 line 702: "Formic acid/CO ratios (which have been used in some studies to determine the contribution of polluted air masses) ranged between $1.0 \times 10^{-3}$ to $2.5 \times 10^{-2}$ ppb ppb$^{-1}$. The ratio peaked consistently in the mid-afternoon, which coincided with when formic acid and CO reached their maximum and minimum, respectively. In addition, there were no spikes in the formic acid/CO ratio during the study, suggesting that contributions of polluted air masses to the daily increase in formic acid are minimal."**

14. Referee's comment: "*line 535: I don't understand comparison of organic ion signal to F2SO2 sensitivity. the ratio here isn't unitless.*"

**Author response:** We acknowledge that the units used are confusing. Hence, we will use the raw ion signals (Hz) of m/z 75, 87, 101, 103, 117 and 131 in the revised manuscript. The revised figures are shown in our reply to comment 1.

15. Referee's comment: "*line 559: what does zeroth order check mean?*"

**Author response:** It means that we checked that the estimated sum of organic carbon contributed by the measured organic acids is consistent with the total gas-phase water-soluble organic carbon ($WSOC_g$). Since the estimated carbon mass fractions of $WSOC_g$ comprised of these organic acids are generally less than 100 %, this suggests that our peak assignments are plausible. To remove any confusion, we have modified the aforementioned sentence in the revised manuscript:

**Page 19 line 602: "This comparison primarily serves as a check to determine if the peak assignments are plausible by ensuring that the estimated sum of organic carbon contributed by these four organic acids is less than or equal to the measured $WSOC_g$."**

16. Referee's comment: "*line 591: How long are these spikes? How close is the nearest power plant or urban area?*"

**Author response:** The $SO_2$ and $HNO_3$ spikes can last between 1 to 3 hours. The closest power plant was Plant Bowen, which was ~25 km north of the site. The closest urban center was Atlanta, which was ~55 km away. This information has been added to the revised manuscript:

**Page 5 line 125: "Briefly, the Yorkville field site (33.931 N, 85.046 W) was located ~55 km northwest of Atlanta (the closest urban center), and was on a broad ridge in a large pasture where there were occasionally grazing cattle. The field site was surrounded by forest and agricultural land. There were no major roads near the field site and nearby traffic emissions were negligible. The closest power plant was Plant Bowen, which was located ~25 km north of the field site."**

**Page 20 line 669: "However, there were occasional periods when the site was impacted by anthropogenic pollution. In particular, there are spikes in both $SO_2$ and $HNO_3$ concentrations lasting between 1 to 3 hours throughout the study that corresponded to the site being impacted by power plant or urban emissions."**

17. Referee's comment: "*line 663: what is a reasonable detection limit? give a number*"

**Author response:** Whether or not a detection limit is reasonable will depend on the species measured. In this study, our detection limits for the organic acids studied (1 to 60 ppt) are typically lower than the concentrations of the organic acids studied, allowing us to be reasonably confident of the concentrations measured. To remove any confusion, we have modified the aforementioned sentence in the revised manuscript:

**Page 23 line 771: "Limits of detection ranged from 1 to 60 ppt for 2.5 min integration periods for the organic acids studied."**

18. Referee's comment: "*Please define all acronyms at first use. I could not find definitions for m/z, slpm.*"

**Author response:** These definitions have been added to the revised manuscript.

[revised manuscript text omitted]

hour averages.

[Figure]

**Figure S3:** Panels (a) and (b) show the sensitivities of formic acid ions (HCOO⁻ at m/z 45,

HCOO⁻•HF at m/z 65, and SF$_4^-$ at m/z 108) obtained from calibration measurements as a function of ambient water vapor and O$_3$ concentrations. Panels (c) and (d) show the acetic acid sensitivity (CH$_3$COO⁻•HF at m/z 79) obtained from calibration measurements as a function of ambient water vapor and O$_3$ concentrations.

[Figure]

**Figure S4:** Example of the CIMS instrument response during switches between background, calibration and ambient measurements of (a) formic, and (c) acetic acids.

Panels (b) and (d) show the percent of formic and acetic acid ion signals after the removal of a 6.75 ppb of formic acid and 5.87 ppb of acetic acid standard addition calibration as a function of time. The data shown here is 13 s time resolution data. Double exponential fits to each m/z ion are shown as colored solid lines. Black dashed lines show the times for the ions to decay to $1/e^2$.

[Figure]

**Figure S5:** Time series and diurnal profiles of ion signals of organic acids with m/z 75, 87, 101, 103, 117 and 131 measured during the field study. The data are displayed as 1-hour averages. All the signals represent averages in 1-hour intervals and the standard errors are plotted as error bars.

Deleted: These organic acids were not calibrated so all the signals are presented here as Hz normalized by the instrument's sensitivity to $F_2{}^{34}SO_2$ (Hz ppb$^{-1}$) which was the primary calibrant used in the field study.

[Figure]

**Figure S6:** Time series of (a) SO₂ and (b) HNO₃ concentrations measured during the field study. All the data are displayed as 1-hour averages.

[Figure]

[Figure]

**Figure S7:** Scatter plots of concentrations (or ion signals) of t̶he measured organic acids with CO concentration. All the data are displayed as 1-hour averages. Red lines shown are linear fits to the data.

[Figure]

**Figure S8:** Scatter plots of concentrations (or ion signals) of the measured organic acids with SO₂ concentration. All the data are displayed as 1-hour averages. Red lines shown are linear fits to the data.

[Figure]

[Figure]

**Figure S9:** Scatter plots of concentrations (or ion signals) of the measured organic acids with $O_3$ concentration. All the data are displayed as 1-hour averages. Red lines shown are linear fits to the data.

[Figure]

[Figure]

[Figure]

**Figure S10:** Scatter plots of concentrations (or ion signals) of the measured organic acids with HNO$_3$ concentration. To exclude periods when the site was affected by urban or power plant emissions, data where HNO$_3$ > 0.5 ppb are excluded from these scatter plots. All the data are displayed as 1-hour averages. Red lines shown are linear fits to the data.

[Figure]

**Figure S11:** (a) Time series of isoprene concentration during the field study. (b) Diurnal profile of isoprene. All the concentrations represent averages in 1-hour intervals and the standard errors are plotted as error bars. (c) Scatter plot of isoprene concentration with ambient temperature. All the data are displayed as 1-hour averages.

[Figure]

**Figure S12:** Scatter plots of concentrations of (a) formic and (b) acetic acids with isoprene concentration. All the data are displayed as 1-hour averages. Red lines shown are linear fits to the data.

---

## Author Response (AR3)

The responses to the comments of the Associate Editor in our direct reply (shown below) and within the revised manuscript (see marked copy) are provided. The pages and lines indicated below correspond to those in the marked copy.

**Response to Associate Editor (Associate Editor's comments are italicized)**

1. Associate editor's comment: "*I partly agree with the anonymous reviewer #1 that the authors should add some discussions on the advantages and disadvantages of SF6- chemistry compared to other ion chemistries that many people use (e.g. Iodide). Although some discussions are shown in the response to reviewers' comments, they are not available yet in the main text. I would encourage the authors to add 1-2 paragraphs either in or before Conclusions for this purpose, even though quantitative comparison may not be possible.*"

**Author response:** As requested, we have added a couple of paragraphs into the revised manuscript pointing out the advantages and disadvantages (specifically, the sensitivities of organic acids) that $SF_6^-$-CIMS has over $I^-$-CIMS:

**Page 13 line 364: "Nevertheless, these sensitivities are compared to formic and acetic acid sensitivities measured by a high-resolution time-of-flight chemical ionization mass spectrometer (Aerodyne Research Inc.) that utilized $I^-$ reagent ions during the field study. Only the formic and acetic acid sensitivities were compared since laboratory calibrations were not performed to determine the sensitivities for oxalic, butyric, glycolic, propionic and valeric acids by $I^-$-CIMS. Although the formic acid sensitivity measured by $I^-$-CIMS ($1.33 \pm 0.28$ Hz ppt$^{-1}$) was comparable to that measured by $SF_6^-$-CIMS, the acetic acid sensitivity measured by $I^-$-CIMS ($< 0.1$ Hz ppt$^{-1}$) was substantially lower than that measured by $SF_6^-$-CIMS. Previous studies have similarly reported low acetic acid sensitivity measured by $I^-$-CIMS (Aljawhary et al., 2013; Lee et al., 2014).**

**Since many recent studies use $I^-$ as a reagent ion to measure many compounds, the measured $SF_6^-$ sensitivities to organic acids are compared with those of $I^-$ reported by Lee et al. (2014, 2018). However, it is important to note that the absolute $SF_6^-$ and $I^-$ sensitivities values are specific to the respective instruments and their configuration. The sensitivity to individual compounds depend on a variety of instrument parameters (e.g., flow rates, pressures, electric fields, ion source activity) that control ion production and transmission, reaction time, declustering efficiency, etc. Consequently, this analysis serves primarily as a qualitative comparison of $SF_6^-$ and $I^-$ sensitivity.**

**Although the $I^-$ sensitivity to formic acid (2.9 Hz ppt$^{-1}$) reported by Lee et al. (2014) is higher than that of $SF_6^-$ (1.29 Hz ppt$^{-1}$), the $SF_6^-$ sensitivities for the other organic acids (i.e., acetic, oxalic, glycolic and propionic acids) are substantially higher than those of $I^-$ (Table S1a). The $SF_6^-$ CIMS method is particularly sensitive to oxalic, propionic and glycolic acids, which are expected to be present at low concentrations in the atmosphere. The sensitivities of $SF_6^-$ and $I^-$ to $SO_2$, $HNO_3$ and HCl can also be compared (Table S1b). The $SF_6^-$ sensitivities of $SO_2$ and HCl are significantly higher than that of $I^-$ reported by Lee et al. (2018). However, $I^-$ is more sensitive to $HNO_3$."**

**Table S1a: Comparison of $SF_6^-$ vs. $I^-$ sensitivities of organic acids**

| Organic Acid | $I^-$ sensitivity (Hz ppt$^{-1}$)[a] | $SF_6^-$ sensitivity (Hz ppt$^{-1}$) | |
|:---:|:---:|:---:|:---:|
| | | $X^-$ | $X^- \cdot HF$ |
| Formic acid | 2.9 | $1.29 \pm 0.22$ | $0.29 \pm 0.05$ |
| Acetic acid | 0.1 | $1.46 \pm 0.29$ | $0.30 \pm 0.06$ |
| Oxalic acid | 0.21 | $6.38 \pm 0.32$ | $0.97 \pm 0.05$ |
| Butyric acid | Not available | $0.41 \pm 0.01$ | $0.12 \pm 0.004$ |
| Glycolic acid | 1.1 | $5.53 \pm 0.11$ | $1.64 \pm 0.03$ |
| Propionic acid | 0.066 | $2.05 \pm 0.02$ | $1.26 \pm 0.01$ |
| Valeric acid | Not available | $0.76 \pm 0.008$ | $0.35 \pm 0.004$ |

[a]The $I^-$ sensitivities shown here are those reported by Lee et al. (2014). The organic acids were detected as cluster ions with iodide ($I(X)^-$).

**Table S1b: Comparison of $SF_6^-$ vs. $I^-$ sensitivities of inorganic compounds**

| Inorganic compound | $I^-$ sensitivity (Hz ppt$^{-1}$)[b] | $SF_6^-$ sensitivity (Hz ppt$^{-1}$) |
|:---:|:---:|:---:|
| $SO_2$ | 0.028 | 2.9 |
| $HNO_3$ | 9.0 | 5.8 for $NO_3^-$, 0.2 for $NO_3^- \cdot HF$[c] |
| HCl | 0.03 | 1.4[d] |

[b]The $I^-$ sensitivities shown here are those reported by Lee et al. (2018).
[c]The high collision energy used in the CDC promoted the dissociation of $NO_3^- \cdot HF$ ions, causing the low sensitivity at $NO_3^- \cdot HF$.
[d]HCl was detected as $SF_5Cl^-$.

**References:**

Aljawhary, D., Lee, A. K. Y., and Abbatt, J. P. D.: High-resolution chemical ionization mass spectrometry (ToF-CIMS): application to study SOA composition and processing, Atmospheric Measurement Techniques, 6, 3211-3224, 10.5194/amt-6-3211-2013, 2013.

Lee, B. H., Lopez-Hilfiker, F. D., Mohr, C., Kurten, T., Worsnop, D. R., and Thornton, J. A.: An Iodide-Adduct High-Resolution Time-of-Flight Chemical-Ionization Mass Spectrometer: Application to Atmospheric Inorganic and Organic Compounds, Environmental Science & Technology, 48, 6309-6317, 10.1021/es500362a, 2014.

Lee, B. H., Lopez-Hilfiker, F. D., Veres, P. R., McDuffie, E. E., Fibiger, D. L., Sparks, T. L., Ebben, C. J., Green, J. R., Schroder, J. C., Campuzano-Jost, P., Iyer, S., D'Ambro, E. L., Schobesberger, S., Brown, S. S., Wooldridge, P. J., Cohen, R. C., Fiddler, M. N., Bililign, S., Jimenez, J. L., Kurtén, T., Weinheimer, A. J., Jaegle, L., and Thornton, J. A.: Flight Deployment of a High-Resolution Time-of-Flight Chemical Ionization Mass Spectrometer: Observations of Reactive Halogen and Nitrogen Oxide Species, Journal of Geophysical Research: Atmospheres, 0, doi:10.1029/2017JD028082, 2018.

2. Associate editor's comment: *"It should be noted that detection of SO2 using I- chemistry was recently demonstrated (Lee et al., 2018). As a result, the sentence "the major advantage that SF6- has over I- and CH3CO2- is that it allows for the detection of acetic acid and SO2". (L110-L111) is not totally accurate. Detection of acetic acid is also possible using I-, but not ideal, as shown in Lee et al., 2014."*

**Author response:** We agree that the Associate Editor has a good point. Lee et al. (2018) did measure $SO_2$ in a nocturnal power plant plume using the $I^-$ reagent ion with a high-resolution TOF mass spectrometer, but they also showed that the sensitivity was approximately 100 times lower than that for formic acid. Hence, we have cited this paper and have revised the manuscript as follows:

[revised manuscript text omitted]
 and $O_3$ concentrations. Data in panels (a) to (d) are displayed as 1-hour averages.

[Figure]

**Figure S3:** Panels (a) and (b) show the sensitivities of formic acid ions (HCOO$^-$ at m/z 45, HCOO$^-\bullet$HF at m/z 65, and SF$_4^-$ at m/z 108) obtained from calibration measurements as a function of ambient water vapor and O$_3$ concentrations. Panels (c) and (d) show the acetic acid sensitivity (CH$_3$COO$^-\bullet$HF at m/z 79) obtained from calibration measurements as a function of ambient water vapor and O$_3$ concentrations.

[Figure]

**Figure S4:** Example of the CIMS instrument response during switches between background, calibration and ambient measurements of (a) formic, and (c) acetic acids. Panels (b) and (d) show the percent of formic and acetic acid ion signals after the removal of a 6.75 ppb of formic acid and 5.87 ppb of acetic acid standard addition calibration as a function of time. The data shown here is 13 s time resolution data. Double exponential fits to each m/z ion are shown as colored solid lines. Black dashed lines show the times for the ions to decay to $1/e^2$.

[Figure]

**Figure S5:** Time series and diurnal profiles of ion signals of organic acids with m/z 75, 87,

101, 103, 117 and 131 measured during the field study. The data are displayed as 1-hour averages. All the signals represent averages in 1-hour intervals and the standard errors are plotted as error bars.

[Figure]

**Figure S6:** Time series of (a) $SO_2$ and (b) $HNO_3$ concentrations measured during the field study. All the data are displayed as 1-hour averages.

[Figure]

**Figure S7:** Scatter plots of concentrations (or ion signals) of the measured organic acids with CO concentration. All the data are displayed as 1-hour averages. Red lines shown are linear fits to the data.

[Figure]

**Figure S8:** Scatter plots of concentrations (or ion signals) of the measured organic acids with $SO_2$ concentration. All the data are displayed as 1-hour averages. Red lines shown are linear fits to the data.

[Figure]

**Figure S9:** Scatter plots of concentrations (or ion signals) of the measured organic acids with $O_3$ concentration. All the data are displayed as 1-hour averages. Red lines shown are linear fits to the data.

[Figure]

**Figure S10:** Scatter plots of concentrations (or ion signals) of the measured organic acids with $HNO_3$ concentration. To exclude periods when the site was affected by urban or power plant emissions, data where $HNO_3 > 0.5$ ppb are excluded from these scatter plots. All the data are displayed as 1-hour averages. Red lines shown are linear fits to the data.

[Figure]

**Figure S11:** (a) Time series of isoprene concentration during the field study. (b) Diurnal profile of isoprene. All the concentrations represent averages in 1-hour intervals and the standard errors are plotted as error bars. (c) Scatter plot of isoprene concentration with ambient temperature. All the data are displayed as 1-hour averages.

[Figure]

**Figure S12:** Scatter plots of concentrations of (a) formic and (b) acetic acids with isoprene
concentration. All the data are displayed as 1-hour averages. Red lines shown are linear
fits to the data.

Table S1a: Comparison of $SF_6^-$ vs. $I^-$ sensitivities of organic acids

| Organic Acid | $I^-$ sensitivity (Hz ppt$^{-1}$)[a] | $SF_6^-$ sensitivity (Hz ppt$^{-1}$) | |
|---|---|---|---|
| | | $X^-$ | $X^-\cdot HF$ |
| Formic acid | 2.9 | $1.29 \pm 0.22$ | $0.29 \pm 0.05$ |
| Acetic acid | 0.1 | $1.46 \pm 0.29$ | $0.30 \pm 0.06$ |
| Oxalic acid | 0.21 | $6.38 \pm 0.32$ | $0.97 \pm 0.05$ |
| Butyric acid | Not available | $0.41 \pm 0.01$ | $0.12 \pm 0.004$ |
| Glycolic acid | 1.1 | $5.53 \pm 0.11$ | $1.64 \pm 0.03$ |
| Propionic acid | 0.066 | $2.05 \pm 0.02$ | $1.26 \pm 0.01$ |
| Valeric acid | Not available | $0.76 \pm 0.008$ | $0.35 \pm 0.004$ |

[a]The $I^-$ sensitivities shown here are those reported by Lee et al. (2014). The organic acids
were detected as cluster ions with iodide ($I(X)^-$).

Table S1b: Comparison of $SF_6^-$ vs. $I^-$ sensitivities of inorganic compounds

| Inorganic compound | $I^-$ sensitivity (Hz ppt$^{-1}$)[b] | $SF_6^-$ sensitivity (Hz ppt$^{-1}$) |
|---|---|---|
| $SO_2$ | 0.028 | 2.9 |
| $HNO_3$ | 9.0 | 5.8 for $NO_3^-$, 0.2 for $NO_3^-\cdot HF$[c] |
| HCl | 0.03 | 1.4[d] |

[b]The $I^-$ sensitivities shown here are those reported by Lee et al. (2018).
[c]The high collision energy used in the CDC promoted the dissociation of $NO_3^-\cdot HF$ ions,
causing the low sensitivity at $NO_3^-\cdot HF$.
[d]HCl was detected as $SF_5Cl^-$.